# Finding the Homology of Decision Boundaries with Active Learning

**Weizhi Li**
Arizona State University
weizhili@asu.edu

**Gautam Dasarathy**
Arizona State University
gautamd@asu.edu

**Karthikeyan Natesan Ramamurthy**
IBM Research
knatesa@us.ibm.com

**Visar Berisha**
Arizona State University
visar@asu.edu

## Abstract

Accurately and efficiently characterizing the decision boundary of classifiers is important for problems related to model selection and meta-learning. Inspired by topological data analysis, the characterization of decision boundaries using their homology has recently emerged as a general and powerful tool. In this paper, we propose an active learning algorithm to recover the homology of decision boundaries. Our algorithm sequentially and adaptively selects which samples it requires the labels of. We theoretically analyze the proposed framework and show that the query complexity of our active learning algorithm depends naturally on the intrinsic complexity of the underlying manifold. We demonstrate the effectiveness of our framework in selecting best-performing machine learning models for datasets just using their respective homological summaries. Experiments on several standard datasets show the sample complexity improvement in recovering the homology and demonstrate the practical utility of the framework for model selection. Source code for our algorithms and experimental results is available at `https://github.com/wayne0908/Active-Learning-Homology`.

## 1 Introduction

A broadly known meta-learning [1] is to design an model to learn a task learning process and, with this model, a new task can be adapted to with fewer examples. [2] views meta-learning from another perspective: The complexity of the data at hand is an important insight that, if gleaned correctly from past experience, can greatly enhance the performance of a meta-learning procedure. A particularly useful characterization of data complexity is to understand the topological properties of the decision boundary; for example, by using topological data analysis (TDA) [3, 4, 5]. This scenario makes sense in settings where large corpora of labeled training data are available to recover the persistent homology of the decision boundary for use in downstream machine learning tasks [5, 6, 7, 8]. However the utility of this family of methods is limited in applications where labeled data is expensive to acquire.

In this paper, we explore the intersection of active learning and topological data analysis for the purposes of efficiently learning the persistent homology of the decision boundary in classification problems. In contrast to the standard paradigm, in active learning, the learner has access to unlabeled data and sequentially selects a set of points for an oracle to label. We propose an efficient active learning framework that adaptively select points for labeling near the decision boundary. A theoretical analysis of the algorithm results in an upper bound on the number of samples required to recover the recover the decision boundary homology. Naturally, this query complexity depends on the intrinsic complexity of the underlying manifold.

There have been several other studies that have explored the use of topological data analysis to characterize the decision boundary in classification problems. In [9], the authors use the persistent homology of the decision boundary to tune hyperparameters in kernel-based learning algorithms. They later extended this work and derived the conditions required to recover the homology of the decision boundary from only samples [5]. Other works have explored the use of other topological features to characterize the difficulty of classification problems [10, 11, 7]. While all previous work assumes full knowledge of data labels, only samples near the decision boundary are used to construct topological features. We directly address this problem in our work by proposing an active approach that adaptively and sequentially labels only samples near the decision boundary, thereby resulting in significantly reduced the complexity of needed labelled samples. To the best of our knowledge, this is the first work that explores the intersection of active learning and topological data analysis.

Our main contributions are as follows:

- We introduce a new algorithm for actively selecting samples to label in service of finding the persistent homology of the decision boundary. We provide theoretical conditions on the query complexity that lead to the successful recovery of the decision boundary homology.
- We evaluate the proposed algorithm for active homology estimation using synthetic data and compare its performance to a passive approach that samples data uniformly. In addition, we demonstrate the utility of our approach relative to a passive approach on a stylized model selection problem using real data.

## 2    Preliminaries

In this section, we define the decision boundary manifold and discuss the labeled Čech Complex [5] which we then use to estimate the homology of this manifold from labeled data. For more background and details, we direct the reader to appendix.

### 2.1    The Decision Boundary Manifold and Data

Let $\mathcal{X}$ be a Euclidean space that denotes the domain/feature space of our learning problem and let $\mu$ denote the standard Lebesgue measure on $\mathcal{X}$. We will consider the binary classification setting in this paper and let $\mathcal{Y} = \{0, 1\}$ denote the label set. Let $p_{XY}$ denote a joint distribution on $\mathcal{X} \times \mathcal{Y}$. Of particular interest to us in this paper is the so-called *Bayes decision boundary* $\mathcal{M} = \{\mathbf{x} \in \mathcal{X} | p_{Y|X}(1|\mathbf{x}) = p_{Y|X}(0|\mathbf{x})\}$. Indeed, identifying $\mathcal{M}$ is equivalent to being able to construct the provably optimal binary classifier called the Bayes optimal predictor:

$$f(\mathbf{x}) = \begin{cases} 1 & \text{if } p_{Y|X}(1 \mid \mathbf{x}) \geq 0.5 \\ 0 & \text{otherwise} \end{cases}. \tag{1}$$

Following along the lines of [5], the premise of this paper relies on supposing that the set $\mathcal{M}$ is in fact a reasonably well-behaved manifold[1]. That is, we will make the following assumption.

**Assumption 1.** *The decision boundary $\mathcal{M}$ is a manifold in $\mathcal{X}$ with a condition number $1/\tau$.*

The condition number $\frac{1}{\tau}$ is an intrinsic property of $\mathcal{M}$ and encodes both the local and global curvature of the manifold. The value $\tau$ is the largest number such that the open normal bundle about $\mathcal{M}$ of radius $r$ is embedded in $\mathbb{R}^K$ for every $r < \tau$. *E.g.*, in Figure 1, $\mathcal{M}$ is a circle in $\mathbb{R}^2$ and $\tau$ is its radius. We refer the reader to the appendix (or [13]) for a formal definition.

Now we will suppose that we have access to $N$ i.i.d samples $\mathcal{D} = \{\mathbf{x}_1, ..., \mathbf{x}_N\} \subset \mathcal{X}$ drawn according to the marginal distribution $p_X$. Notice that in a typical passive learning setting, we assume access to the $N$ corresponding labels as well. The goal of this paper is to demonstrate that if we are allowed to choose the labels observed in a sequential and adaptive fashion, then we may obtain far fewer than $N$ labels while still being competitive with the traditional passive learning approaches. Based on the observed data, we define the set $\mathcal{D}^0 = \{\mathbf{x} \in \mathcal{D} : f(\mathbf{x}) = 0\}$, that is the set of all samples with Bayes optimal label of 0; similarly, we let $\mathcal{D}^1 = \{\mathbf{x} \in \mathcal{D} : f(\mathbf{x}) = 1\}$.

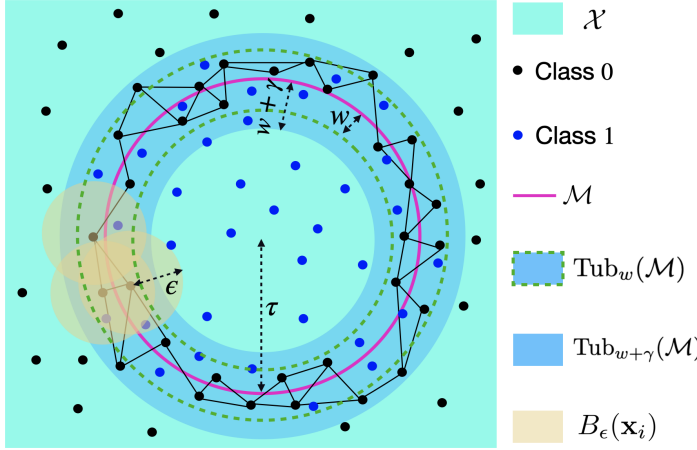

**Figure 1:** An example of $(\epsilon, \gamma)-$labeled Čech complex, constructed in a tubular neighborhood $\mathrm{Tub}_{w+\gamma}(\mathcal{M})$ of radius $w + \gamma$, for a manifold $\mathcal{M}$ of condition number $1/\tau$. The overlap between the two classes $(\mathfrak{D})$ is contained in $\mathrm{Tub}_w(\mathcal{M})$. The complex is constructed on samples in class $0$, by placing balls of radius $\epsilon$ $(B_\epsilon(\mathbf{x}_i))$, and is "witnessed" by samples in class $1$. $\mathcal{X}$ is the compact probability space for the data. Each triangle is assumed to be a $2-$simplex in the simplicial complex. Note that we keep the samples from both classes sparse for aesthetic reasons.

Legend:
- $\mathcal{X}$
- Class 0
- Class 1
- $\mathcal{M}$
- $\mathrm{Tub}_w(\mathcal{M})$
- $\mathrm{Tub}_{w+\gamma}(\mathcal{M})$
- $B_\epsilon(\mathbf{x}_i)$

## 2.2 The Labeled Čech Complex

As outlined in Section 1, our goal is to recover the homological summaries of $\mathcal{M}$ from data. Homological summaries such as Betti numbers estimate the number of connected components and the number of holes of various dimensions that are present in $\mathcal{M}$. Since we only have a sample of data points in practice, we first construct a simplicial complex from these points that mimics the shape of $\mathcal{M}$. We can then estimate the rank of any homology group $H_i$ of dimension $i$ from this complex. This rank is called the Betti number $\beta_i$ and informally denotes the number of holes of dimension $i$ in the complex. The multi-scale estimation of Betti numbers, which involves gradual "thickening" of the complex, results in a persistence diagram $\mathrm{PD}_i$ that encodes the *birth* and *death time* of the $i-$dimensional holes in the complex. For more background we refer the reader to [14].

In [5], the authors consider the passive learning setting for estimating the homology of $\mathcal{M}$ and propose a simplicial approximation for the decision boundary called the Labeled Čech (LČ) Complex. We now provide a definition of this complex, letting $B_\epsilon(\mathbf{x})$ denote a ball of radius $\epsilon$ around $\mathbf{x}$. We refer the reader to the appendix or [5] for the more details.

**Definition 1.** *Given $\epsilon, \gamma > 0$, an $(\epsilon, \gamma)$-labeled Čech complex is a simplicial complex constructed from a collection of simplices such that each simplex $\sigma$ is formed on the points in a set $S \subseteq \mathcal{D}^0$ witnessed by the reference set $\mathcal{D}^1$ satisfying the following conditions: (a) $\bigcap_{\mathbf{x}_i \in \sigma} B_\epsilon(\mathbf{x}_i) \neq \emptyset$, where $\mathbf{x}_i \in S$ are the vertices of $\sigma$. (b) $\forall \mathbf{x}_i \in S \subseteq \mathcal{D}^0, \exists \mathbf{x}_j \in \mathcal{D}^1$ such that, $\|\mathbf{x}_i - \mathbf{x}_j\|_2 \leq \gamma$.*

The set $S$ is used to construct the LČ complex witnessed by the reference set $\mathcal{D}^1$. This allows us to infer that each vertex of the simplices $\sigma$ are within distance $\gamma$ to some point in $\mathcal{D}^1$. The authors in [5] show that, under certain assumption on the manifold and the distribution, provided sufficiently many random samples (and their labels) drawn according $p_{XY}$, the set $U = \bigcup_{\mathbf{x}_i \in \sigma} B_\epsilon(\mathbf{x}_i)$ forms a cover of $\mathcal{M}$ and deformation retracts to $\mathcal{M}$. Moreeover, the nerve of the covering is homotopy equivalent to $\mathcal{M}$. The assumptions under which the above result holds also turns out to be critical to achieve the results of this paper, and hence we will devote the rest of this section to elaborating on these.

Before stating our assumptions, we need a few more definitions. For the distribution $p_{XY}$, we will let

$$\mathfrak{D} \triangleq \{\mathbf{x} \in \mathcal{X} : p_{X|Y}(\mathbf{x} \mid 1) p_{X|Y}(\mathbf{x} \mid 0) > 0\}.$$

In other words, $\mathfrak{D}$ denotes the region of the feature space where both classes overlap, i.e., both class conditional distributions $p_{X|Y}(\cdot \mid 1)$ and $p_{X|Y}(\cdot \mid 0)$ have non-zero mass. For any $r > 0$, we let $\mathrm{Tub}_r(\mathcal{M})$ denote a "tubular" neighborhood of radius $r$ around $\mathcal{M}$ [13]. We will let $\mathrm{Tub}_w(\mathcal{M})$ denote the smallest tubular neighborhood of $\mathcal{M}$ that encloses $\mathfrak{D}$. That is

$$w \triangleq \arg\inf \{\rho > 0 : \mathrm{Tub}_\rho(\mathcal{M}) \supset \mathfrak{D}\}.$$

A stylized example that highlights the relationship between these parameters is shown in Figure 1. In this sequel, we will introduce a relevant assumption and a lemma underlying the results of this paper

(similar to those in [13, 5]). All assumptions and follow-on results are dependent on two parameters that are specific to the joint distribution: $w$ - the amount of overlap between the distributions and $\tau$ - the global geometric property of the decision boundary manifold.

**Assumption 2.** $w < (\sqrt{9} - \sqrt{8})\tau$.

A principal difference between [13] and our work is that [13] has direct access to the manifold and supposes that all generated samples are contained within $\mathrm{Tub}_{(\sqrt{9}-\sqrt{8})\tau}(\mathcal{M})$; this is one of the sufficient conditions for manifold reconstruction from samples. In contrast, as in [5], we do not have direct access to the decision boundary manifold. Rather, certain properties of $\mathcal{M}$ are *inferred* from the labels. In this paper, we show that we can infer the homology of $\mathcal{M}$ with far fewer labels if we are allowed to sequentially and adaptively choose which labels to obtain.

Since the generated samples do not necessarily reside within $\mathrm{Tub}_{(\sqrt{9}-\sqrt{8})\tau}(\mathcal{M})$, it is not immediately apparent that it is possible to find an $S$ (see Definition 1) that is entirely contained in $\mathrm{Tub}_{(\sqrt{9}-\sqrt{8})\tau}(\mathcal{M})$. However, Assumption 2 allows us to guarantee precisely this. To see this, we will first state the following lemma.

**Lemma 1.** *Provided $\mathcal{D}^0$ and $\mathcal{D}^1$ are both $\frac{\gamma}{2}$-dense in $\mathcal{M}$, then $S$ is contained in $Tub_{w+\gamma}(\mathcal{M})$ and it is $\frac{\gamma}{2}$-dense[2] in $\mathcal{M}$.*

The lemma 1 is based on definition 1(b) that $S$ is a subset of $\mathcal{D}^0$ within distance $\gamma$ from $\mathcal{D}^1$ and the proof goes as follows. $\mathcal{D}^0$ and $\mathcal{D}^1$ being $\frac{\gamma}{2}$-dense in $\mathcal{M}$ implies $\|\mathbf{x}_i - \mathbf{x}_j\|_2 < \gamma$, where $\mathbf{x}_i \in \mathcal{D}^0 \bigcap B_{\frac{\gamma}{2}}(\mathbf{p})$ and $\mathbf{x}_j \in \mathcal{D}^1 \bigcap B_{\frac{\gamma}{2}}(\mathbf{p})$ for every $\mathbf{p} \in \mathcal{M}$. Therefore, as the distance from $\mathbf{x}_i \in S \subseteq \mathcal{D}^0$ to $\mathbf{x}_j \in \mathcal{D}^1$ is bounded by $\gamma$, $S$ is also $\frac{\gamma}{2}$-dense in $\mathcal{M}$. This immediately implies $S$ is $(w + \gamma)$-dense in $\mathcal{M}$. Given lemma 1, as our initial sampling strategy is such that both $\mathcal{D}^0$ and $\mathcal{D}^1$ are $\frac{\gamma}{2}$-dense in $\mathcal{M}$, $S$ is contained in $Tub_{w+\gamma}(\mathcal{M})$ and $\frac{\gamma}{2}$-dense in $\mathcal{M}$. and provided we choose $\gamma < (\sqrt{9} - \sqrt{8})\tau - w$, we can guarantee that $S \subset \mathrm{Tub}_{(\sqrt{9}-\sqrt{8})\tau}(\mathcal{M})$.

With $S \subset \mathrm{Tub}_{(\sqrt{9}-\sqrt{8})\tau}(\mathcal{M})$, $S$ being $(w + \gamma)$-dense in $\mathcal{M}$ and an appropriate $\epsilon$ properly selected, we will have the $(\epsilon, \gamma)$-LČ complex homotopy equivalent to $\mathcal{M}$ per theorems in [5, 13]. We now state the following proposition.

**Proposition 1.** *The $(\epsilon, \gamma)$-LČ complex is homotopy equivalent to $\mathcal{M}$ as long as (a) $\gamma < (\sqrt{9} - \sqrt{8})\tau - w$; (b) $\mathcal{D}^0$ and $\mathcal{D}^1$ are $\frac{\gamma}{2}$-dense in $\mathcal{M}$ and (c) $\epsilon \in \left( \frac{(w+\gamma+\tau) - \sqrt{(w+\gamma)^2+\tau^2-6\tau(w+\gamma)}}{2}, \frac{(w+\gamma+\tau) + \sqrt{(w+\gamma)^2+\tau^2-6\tau(w+\gamma)}}{2} \right).$*

Notice that the range for $\epsilon$ in (c) above is non-trivial as long as $\gamma < (\sqrt{9} - \sqrt{8})\tau - w$. A pictorial description of relations between $w$, $\gamma$ and $\tau$ is shown in the stylized example in Figure 1; here $\mathcal{M}$ is a circle, $\mathrm{Tub}_{w+\gamma}(\mathcal{M})$ is an annulus and the radius $\epsilon$ of the covering ball $B_\epsilon(\mathbf{x})$ is constrained by $\tau$.

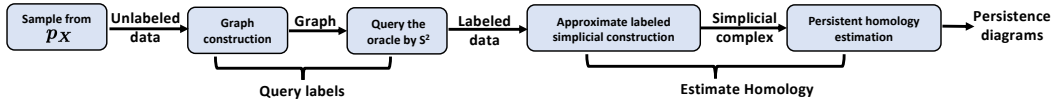

**Figure 2:** The proposed active learning framework for finding the homology of decision boundaries.

## 3 Active Learning for Finding the Homology of Decision Boundaries

As the definitions above and results from [5] make clear, constructing a useful LČ complex requires sampling both class-conditional distributions in the region around the decision boundary to a sufficient resolution. The key insight of our paper is to devise a framework based on active learning that sequentially and adaptively decides where to obtain data and therefore query-efficiently samples points near the decision boundary. In what follows, we will provide a brief description of our algorithm, and then we will establish rigorous theoretical guarantees on the query complexity of the proposed algorithm.

## 3.1 The Active Learning Algorithm

A schematic diagram of the proposed active learning framework is presented in Figure 2. As illustrated, the framework starts from sampling sufficient unlabelled data from $p_X$. Subsequently, the framework takes as input an unlabeled dataset $\mathcal{D}$, and this dataset is used to generate an appropriate graph on the data. This graph is then used to iteratively query labels near the decision boundary. The subset of labeled samples are used to estimate the homology, resulting in the persistence diagram of the LČ complex. We briefly outline the label query and homology estimation phases below, and refer the reader to the appendices for the full details.

**Label query phase:** The label query phase starts with constructing a graph $G = (\mathcal{D}, E)$ from the unlabeled dataset $\mathcal{D}$. While other choices are possible, we will suppose that the graph we construct is either a $k$-radius nearest neighbor or a $k$-nearest neighbors graph[3]. After graph construction, a graph-based active learning algorithm ($S^2$) path [15] accepts the graph $G = (\mathcal{D}, E)$ and selects the data points whose labels it would like to see. This selection is based on the structure of the graph and all previous gathered labels. Specifically, $S^2$ continually queries for the label of the vertex that bisects the shortest path between any pair of oppositely-labeled vertices. The authors in [15] show that $S^2$ can provably query and efficiently locate the cut-set in this graph (i.e., the edges of the graph that have oppositely labeled vertices). As a result, the query phase outputs a set $\tilde{\mathcal{D}}$ associated with the labels that is near the decision boundary.

**Homology estimation phase:** During the homology estimation stage, we construct an approximation of the LČ complex from the query set $\tilde{\mathcal{D}}$. Specifically, we construct the locally scaled labeled Vietoris-Rips (LS-LVR) complex introduced in [5]. Sticking to the query set $\tilde{\mathcal{D}}$ as an example, there are two steps to construct the LS-LVR complex: (1) Generate an initial graph from $\tilde{\mathcal{D}}$ by creating an edge set $\tilde{E}$ as follows: $\tilde{E} = \{\{\mathbf{x}_i, \mathbf{x}_j\} | (\mathbf{x}_i, \mathbf{x}_j) \in \tilde{\mathcal{D}}^2 \wedge y_i \neq y_j \wedge \|\mathbf{x}_i - \mathbf{x}_j\| \leq \kappa \sqrt{\rho_i \rho_j}\}$. Here, $\kappa$ is a scale parameter, $\rho_i$ is the smallest radius of a sphere centered at $\mathbf{x}_i$ to enclose $k$-nearest opposite class neighbors and $\rho_j$ has a similar definition. This creates a bipartite graph where every edge connects points from opposite classes; (2) Connect all 2-hop neighbors to build a simplicial complex. Varying scale parameter $\kappa$ produces a filtration of the simplicial complex, which can be used to estimate persistence diagrams that quantify the persistent homology.

## 3.2 Query Complexity of the Active Learning Algorithm

Let $G = (\mathcal{D}, E)$ denote a $k$-radius neighbor graph constructed from the dataset $\mathcal{D}$. This allows us to define the cut-set $C = \{(\mathbf{x}_i, \mathbf{x}_j) | y_i \neq y_j \wedge (\mathbf{x}_i, \mathbf{x}_j) \in E\}$ and cut-boundary $\partial C = \{\mathbf{x} \in V : \exists e \in C \text{ with } \mathbf{x} \in e\}$. We begin by sketching a structural lemma about the graph $G$ and refer the reader to the appendix for a full statement and proof.

**Lemma 2.** *Suppose $\mathcal{D}^0$ and $\mathcal{D}^1$ are $\frac{\gamma}{2}$-dense in $\mathcal{M}$, then the graph $G = (\mathcal{D}, E)$ constructed from $\mathcal{D}$ is such that $\mathcal{D}^0 \bigcap \partial C$ and $\mathcal{D}^1 \bigcap \partial C$ are both $\frac{\gamma}{2}$-dense in $\mathcal{M}$ and $\partial C \subseteq \text{Tub}_{w+\gamma}(\mathcal{M})$ for $k = \gamma$.*

$\mathcal{D}^0$ and $\mathcal{D}^1$ being $\frac{\gamma}{2}$-dense in $\mathcal{M}$ indicates the longest distance between $\mathbf{x}_i \in \mathfrak{D}^0 \bigcap B_{\frac{\gamma}{2}}(\mathbf{p})$ and $\mathbf{x}_j \in \mathfrak{D}^0 \bigcap B_{\frac{\gamma}{2}}(\mathbf{p})$ for $\mathbf{p} \in \mathcal{M}$ is $\gamma$. Therefore, letting $k = \gamma$ as Lemma 2 suggested results in both $\mathcal{D}^0 \bigcap \partial C$ and $\mathcal{D}^1 \bigcap \partial C$ being $\frac{\gamma}{2}$-dense in $\mathcal{M}$. Similar to Lemma 1, constructing a graph with a $\gamma$ radius inevitably results in a subset of points in $\partial C$ leaking out of $\text{Tub}_w(\mathcal{M})$ and we formally have $\partial C \subseteq \text{Tub}_{w+\gamma}(\mathcal{M})$. The key intuition behind our approach is that $S^2$ is naturally turned to focusing the labels acquired within $\text{Tub}_{w+\gamma}(\mathcal{M})$. As we show below, this is done in a remarkably query efficient manner, and furthermore, when we obtain labeled data via querying we can construct an LČ complex; this allows us to find the homology of the manifold $\mathcal{M}$. We next need some structural assumptions about the manifold $\mathcal{M}$.

**Assumption 3.** *(a)* $\inf_{\mathbf{x} \in \mathcal{M}} \mu_{\mathcal{X}|y}(B_{\gamma/4}(\mathbf{x})) > \rho^y_{\gamma/4}, y \in \{0, 1\}$. *(b)* $\sup_{\mathbf{x} \in \mathcal{M}} \mu_{\mathcal{X}}(B_{(w+\gamma)}(\mathbf{x})) < h_{(w+\gamma)}$. *(c)* $\mu_{\mathcal{X}}(\text{Tub}_{w+\gamma}(\mathcal{M})) \leq N_{w+\gamma} h_{w+\gamma}$.

Assumption 3(a) ensures sufficient mass in both classes such that $\mathcal{D}^0$ and $\mathcal{D}^1$ are $\frac{\gamma}{2}$-dense in $\mathcal{M}$. Assumption 3(b)(c) upper-bounds the measure of $\text{Tub}_{w+\gamma}(\mathcal{M})$. Recall that $G = (\mathcal{D}, \tilde{E})$ in Lemma 2

is a labeled graph; we further write $\beta$ to denote the proportion of the smallest connected component with all the examples identically labeled. We lay out our main theorem as follows

**Theorem 1.** *Let $N_{w+\gamma}$ be the covering number of the manifold $\mathcal{M}$. Under Assumptions 1, 2 and 3, for any $\delta > 0$, we have that the $(\epsilon, \gamma)$-LČ complex estimated by our framework is homotopy equivalent to $\mathcal{M}$ with probability at least $1 - \delta$ provided*

$$|\tilde{\mathcal{D}}| > \frac{\log\left\{1/\left[\beta\left(1 - \sqrt{1 - \delta}\right)\right]\right\}}{\log\left[1/(1 - \beta)\right]} + |\mathcal{D}|N_{w+\gamma}h_{w+\gamma}(\lceil\log_2|\mathcal{D}|\rceil + 1) \tag{2}$$

*where*

$$|\mathcal{D}| > \max\left\{\frac{1}{P(y = 0)\rho_{\gamma/4}^0}\left[\log\left(2N_{\gamma/4}\right) + \log\left(\frac{1}{(1 - \sqrt{1 - \delta})}\right)\right],\right.$$
$$\left.\frac{1}{P(y = 1)\rho_{\gamma/4}^1}\left[\log\left(2N_{\gamma/4}\right) + \log\left(\frac{1}{(1 - \sqrt{1 - \delta})}\right)\right]\right\} \tag{3}$$

**Remark 1.** *Theorem 1 demonstrates that our active learning framework has a query complexity of $\mathcal{O}(NN_{w+\gamma}h_{w+\gamma}log_2N)$. That is, after $\mathcal{O}(NN_{w+\gamma}h_{w+\gamma}log_2N)$ queries at most, a $(\epsilon, \gamma) - $ LČ complex constructed from the queried examples will be homotopy equivalent to $\mathcal{M}$ with high probability. Notice that the intrinsic complexity of the manifold naturally plays a significant role, and the more complex the manifold the more significant gains the active learning framework has over its passive counterpart (cf. Eq. 3). In the appendix, we also provide a simple and concrete example that numerically shows the improvement in query complexity associated with our proposed framework relative to its passive counterpart.*

**Remark 2.** *The results of Theorem 1 can be improved by carrying out a more intricate analysis of the active learning algorithm as in [15]. Indeed, one may also replace the $S^2$ algorithm in our framework with a different graph-based active learning algorithm seamlessly to leverage the properties of that algorithm for active homology estimation of decision boundaries. These, and the relaxation of Assumption 2, are promising directions for future work.*

**Remark 3.** *Parameters $w$ and $\tau$ are intrinsic properties of $p_{xy}$ and $\mathcal{M}$ and these properties are fixed to a classification problem. Variables $\gamma$ and $\epsilon$ are algorithm variables and they are bounded as stated in Proposition 1.*

We provide a complete proof of Theorem 1 in the appendix. However, we will provide some intuition about the operation of our algorithm, and hence to the proof of the theorem here.

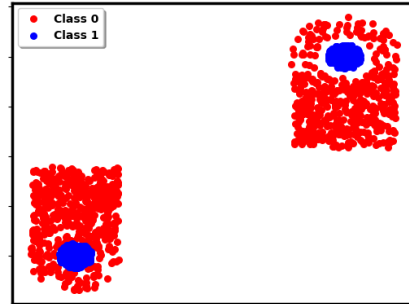

**Figure 3:** Visualization of the synthetic data.

The $S^2$ algorithm is split into two phases: uniform querying of labels and querying via path bisection. The uniform querying serves to finding a path connecting vertices of opposite labels. The path bisection phase queries at the mid-point of the shortest path that connects oppositely labeled vertices in the underlying graph. As the authors in [15] show, this endows $S^2$ with the ability to quickly narrow in on the cut-boundary $\partial C$. The uniform querying phase accounts for the first term in Eq. 2, which guarantees that there are sufficient paths to identify $\partial C$ completely.

During the path bisection phase, we take $(\lceil\log_2|\mathcal{D}|\rceil + 1)$ queries at most (this may be tightened using the techniques in [15]) to find the end point of the cut-edge inside a path; this needs to be done at most $|\partial C|$ to complete the querying phase. Next, with the Assumption 2 and the Lemma 2, it is guaranteed that $\partial C \subseteq \text{Tub}_{w+\gamma}(\mathcal{M}) \subset \text{Tub}_{(\sqrt{9}-\sqrt{8})\tau}(\mathcal{M})$ with $\gamma$ properly selected following Proposition 1. Therefore, we may use the measure $N_{w+\gamma}h_{w+\gamma}$ from Assumption 3(b)(c) to upper-bound $|\partial C|$ which results in the second term of Eq. 2. This naturally ties the query complexity to the manifold complexity via $N_{w+\gamma}$ and $\text{Tub}_{w+\gamma}(\mathcal{M})$. Eq. 3 comes from the necessary condition for the LČ complex being homotopy equivalent to $\mathcal{M}$, following the lines of [5].

# 4 Experimental Results

We compare the homological properties estimated from our active learning algorithm to a passive learning approach on both synthetic data and real data.

In the experiments we use the characteristics of homology group of dimension 1 ($\beta_1$, PD$_1$). We chose to use dimension 1 since [5] shows that this provides the best topological summaries for applications related to model selection. We have the performance evaluation for using the characteristics of homology group of dimension 0 ($\beta_0$, PD$_0$) presented in the appendix.

Using the synthetic data, we study the query complexity of active learning by examining the homological summaries $\beta_1$ and PD$_1$. For real data, we estimate PD$_1$ of the Banknote, MNIST and CIFAR10 and then utilize PD$_1$ to do model selection from several families of classifiers.

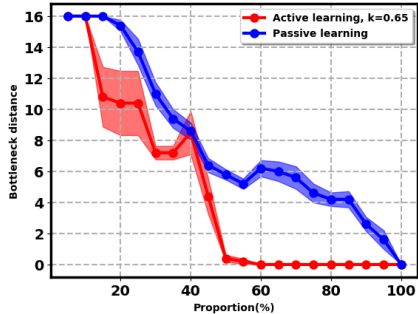

## 4.1 Experiments on Synthetic Data

The synthetic data in Figure 3 has decision boundaries that are homeomorphic to two disjoint circles. This dataset has 2000 examples. Clearly from Figure 3, $\beta_1$ of the decision boundary is two.

Per the first step of our active learning algorithm, we construct a $k$-radius NN graph with $k = 0.65$. The scale parameter is set assuming we have full knowledge of the decision boundary manifold. Subsequently, we use $S^2$ to query the labels of examples on the created graph. After the label query phase, we construct the LS-LVR complex with the queried samples and compute $\beta_1$ and PD$_1$ using

**Figure 4:** Bottleneck distance from ground-truth PD$_1$ by the passive learning and active learning.

the python wrapper of the Ripser package [16, 17]. For the passive learning baseline, we uniformly query the examples with all other aspects of the experiment remaining identical to the active case. We also compute $\beta_1$ and PD$_1$ from the complete dataset and consider them as the "ground-truth" homology summaries. We evaluate the similarities between the estimated homology summaries and the ground-truth homology summaries to show the effectiveness of our active learning framework.

We compare the bottleneck distance [18, 19] between the ground-truth and estimated values of PD$_1$ for different percentages of data labelling. These results are shown on Figure 4. As is clear from the figure, the bottleneck distance for our active learning framework decreases faster than the passive learning approach and perfectly recovers the homology with only 50% of data. A visualization of the query process is shown on Figure 5. As expected, the active learning framework selects more examples to query near the decision.

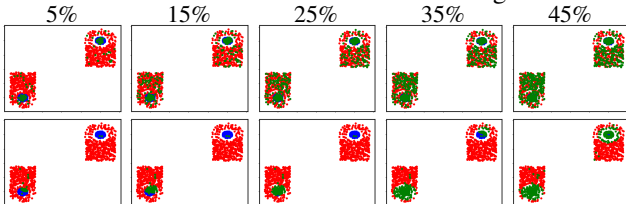

**Figure 5:** Visualization of the query process by passive learning (top row) and our active learning framework (bottom row) for different percentages of data labelling. More examples (highlighted by green) near the decision boundaries are selected to query in the proposed framework.

Please refer to the appendix to evaluate the performance of the active learning framework for different $k$-radius nearest neighbor graphs and $\beta_1$ recovery.

## 4.2 Experiments on Real Data

To demonstrate the effectiveness of our active learning framework on real data, we consider the classifier selection problem discussed in [5]. A bank of pretrained classifiers is accessible in the marketplace and customers select a proper one without changing the hyperparameters of the selected classifier. We consider two selection strategies as follow. One is topologically-based where the classifier with the smallest bottleneck distance from PD$_1$ of queried data is selected. The other one is to ensemble the topologically-selected classifier and the classifier selected based on the validation error of the queried data. We ensemble these two classifiers by averaging the output probabilities.

| Banknote | KNN | SVM | Neural network | Decision tree |
|---|---|---|---|---|
| Passive | $0.1072\pm0.0000$ | $0.3753\pm0.0005$ | $0.4316\pm0.0000$ | $0.1997\pm0.0000$ |
| Active[1] | $0.0783\pm0.0014$ | $0.3231\pm0.0012$ | $0.4316\pm0.0000$ | $0.1901\pm0.0004$ |
| Active[2] | $0.1017\pm0.0001$ | $0.3431\pm0.0012$ | $0.3730\pm0.0138$ | $0.1744\pm0.0026$ |
| Active[3] | **$0.0346\pm0.0013$** | **$0.0836\pm0.0133$** | **$0.1058\pm0.0265$** | **$0.1613\pm0.0004$** |
| Passive (ens) | $0.0176\pm0.0000$ | **$0.0259\pm0.0000$** | $0.0068\pm0.0000$ | $0.0741\pm0.0000$ |
| Active[1] (ens) | $0.0173\pm0.0000$ | **$0.0259\pm0.0000$** | **$0.0039\pm0.0000$** | **$0.0731\pm0.0000$** |
| Active[2] (ens) | **$0.0149\pm0.0000$** | **$0.0259\pm0.0000$** | $0.0134\pm0.0001$ | **$0.0731\pm0.0000$** |
| Active[3] (ens) | **$0.0149\pm0.0000$** | **$0.0259\pm0.0000$** | $0.0072\pm0.0000$ | $0.0770\pm0.0000$ |
| **MNIST** | KNN | SVM | Neural network | Decision tree |
| Passive | $0.0129\pm0.0000$ | **$0.0141\pm0.0000$** | $0.0202\pm0.0000$ | **$0.0332\pm0.0000$** |
| Active[1] | $0.0128\pm0.0000$ | $0.0161\pm0.0001$ | **$0.0150\pm0.0000$** | $0.0388\pm0.0001$ |
| Active[2] | $0.0122\pm0.0000$ | $0.0162\pm0.0001$ | $0.0177\pm0.0000$ | **$0.0332\pm0.0000$** |
| Active[3] | **$0.0104\pm0.0000$** | $0.0156\pm0.0001$ | $0.0388\pm0.0020$ | **$0.0332\pm0.0000$** |
| Passive (ens) | $0.0119\pm0.0000$ | $0.0124\pm0.0000$ | **$0.0104\pm0.0000$** | $0.0290\pm0.0000$ |
| Active[1] (ens) | $0.0123\pm0.0000$ | **$0.0119\pm0.0000$** | **$0.0104\pm0.0000$** | $0.0284\pm0.0000$ |
| Active[2] (ens) | $0.0108\pm0.0000$ | **$0.0119\pm0.0000$** | $0.0125\pm0.0000$ | $0.0284\pm0.0000$ |
| Active[3] (ens) | **$0.0104\pm0.0000$** | **$0.0119\pm0.0000$** | $0.0127\pm0.0000$ | **$0.0274\pm0.0000$** |
| **CIFAR10** | KNN | SVM | Neural network | Decision tree |
| Passive | **$0.3065\pm0.0002$** | $0.4683\pm0.0000$ | $0.3185\pm0.0000$ | **$0.3625\pm0.0000$** |
| Active[1] | $0.3201\pm0.0000$ | $0.4591\pm0.0005$ | **$0.3058\pm0.0006$** | **$0.3625\pm0.0000$** |
| Active[2] | $0.3095\pm0.0001$ | **$0.4007\pm0.0038$** | **$0.3058\pm0.0006$** | **$0.3625\pm0.0000$** |
| Active[3] | $0.3109\pm0.0001$ | $0.4464\pm0.0005$ | $0.3185\pm0.0000$ | **$0.3625\pm0.0000$** |
| Passive (ens) | $0.2987\pm0.0001$ | $0.2698\pm0.0000$ | $0.2651\pm0.0001$ | **$0.3137\pm0.0002$** |
| Active[1] (ens) | $0.2911\pm0.0001$ | $0.2797\pm0.0000$ | **$0.2558\pm0.0000$** | $0.3146\pm0.0000$ |
| Active[2] (ens) | $0.2987\pm0.0001$ | $0.2864\pm0.0003$ | $0.2649\pm0.0001$ | $0.3214\pm0.0005$ |
| Active[3] (ens) | **$0.2935\pm0.0000$** | **$0.2665\pm0.0000$** | $0.2615\pm0.0001$ | $0.3221\pm0.0004$ |

**Table 1:** Average test error rates(five trials) on Banknote, MNIST and CIFAR10 for the model selected with 15% unlabelled pool data. Passive/Active stands for the non-ensemble classifiers selected by the $\mathbf{PD}_1$ homological similarities. Passive/Active (ens) stands for the classifiers ensembled from two classifiers: one is selected by the $\mathbf{PD}_1$ homological similarities and the other one is selected by the validation error. The subscript 1, 2 and 3 of the active learning indicates the used 3NN, 5NN and 7NN graphs. Best performance in the non-ensemble and ensemble cases are boldfaced.

We split the data to a training set, a test set and an unlabeled data pool. The training set is used to generate four different banks of classifiers: $k$-NN with $k$ ranging from 1 to 29, SVM with polynomial kernel function degree ranging from 1 to 14, decision tree with maximum depth ranging from 1 to 27, and neural networks with the number of layers ranging from 1 to 6. The test set is used to evaluate the test error of each classifier. The unlabelled data pool is used to evaluate our active learning algorithm via selective querying.

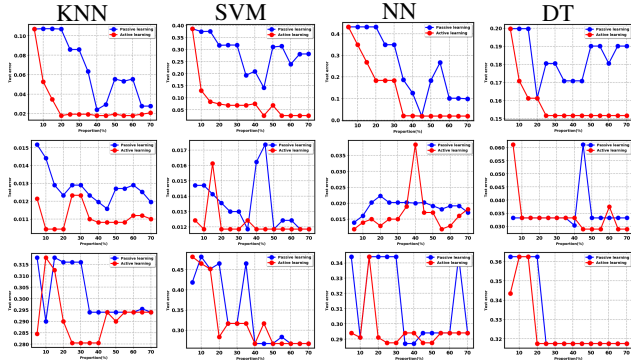

**Figure 6:** Test errors as a function of proportions of queried data on banknote (top), MNIST (middle) and CIFAR10 (bottom) in the model selection (non-ensemble) for different classifier families.

We use the proposed active learning framework to estimate the homological properties of the queried data: constructing a $k$-nearest neighbors graph, query examples by $S^2$ and computing the $\text{PD}_1$ with the queried examples. We set $k = 3, 5,$ and 7. For passive learning, we keep all the operations the same as the active learning framework except the queried examples are collected by uniform random sampling. To compute the PD of the decision boundary of the classifier, we simply use the test set input and the classifier output. Having estimated the homological summaries from the queried data and the classifiers, we compute the bottleneck distance between the $\text{PD}_1$ of the queried data and the classifiers. For the non-ensemble method, we simply select the classifier having the smallest bottleneck distance as a topologically-selected classifier. For the ensemble method, we further include

an additional classifier selected based on the validation error computed from the queried data and ensemble it with the topologically-selected classifier.

We implement the above procedure and evaluate on Banknote [20], MNIST [21] and CIFAR10 [22] datasets. Banknote contains 1372 instances in two classes with four input features for a binary classification task. We randomly sample 100 examples to construct the training set. Given the small size of the Banknote dataset, we use the remaining data as both the test set and unlabelled data pool. Although the test set and the data pool are not rigorously split in the Banknote case, it is still a fair comparison since the performance difference is only subject to the querying strategy. For the MNIST and the CIFAR10 datasets, we create (1) a 1 vs. 8 classification task from MNIST and (2) an automobile vs. ship classification task from CIFAR10. We randomly sample the data to create a training set with a sample size of 200, a test set with a sample size of 2000, and an unlabelled data pool with a sample size of 2000.

Table 1 shows the test error on banknote, MNIST and CIFAR10 for the classifier selected by querying 15% of the unlabelled data pool. We observe that the classifiers selected by our proposed active learning framework generally has a lower test error rate than the passive learning, especially in an ensemble classifier selection framework. As the experimental set-ups are identical (except for the querying strategy) we attribute the performance improvement to the proposed active learning framework used during model selection. Figure 6 indicates the performance of the non-ensemble classifiers selected by the homological similarities at the cost of the different proportions of the data pool. As expected, the proposed active learning framework achieves the best model selection faster than the passive learning for all classifier families. Note that the selection performance may be unstable with an increasing number of the queries since the active learning algorithm exhausts informative examples rapidly and begins to query noisy examples. In summary, Table 1 and Figure 6 indicate that the advantage of active learning in finding good homology summaries is also useful for model selection; this is evidenced by the lower error rates for the active learning approach relative to the passive learning approach.

## 4.3 Analysis of Homological Properties of Real Data

We present the homological properties estimated by passive learning and the proposed active learning framework. Similar to the experiments with the synthetic dataset, we access the complete unlabelled data pool and their labels to compute $\beta_1$ and $PD_1$ and use them as the ground-truth $\beta_1$ and $PD_1$. We then query the unlabelled data pool and estimate $\beta_1$ and $PD_1$ from the queried data. As we observe in the Figure 7(a), $\beta_1$ estimated by our active learning algorithm has a more similar trend to the ground-truth $\beta_1$ in all three real datasets. Furthermore, CIFAR10 has a significantly higher $\beta_1$ than MNIST and Banknote datasets indicating more complex decision boundaries. This is consistent with the Table 1 which shows that the error rates for the CIFAR10 binary classification tasks is higher than the other two datasets. Figure 7(b) shows the bottleneck distance between the estimated $PD_1$ and the ground-truth $PD_1$ for different proportions of labelled data. We observe that the active learning algorithm maintains a smaller bottleneck distance at early stages of querying. Such benefits gradually diminish as more of the data is labelled.

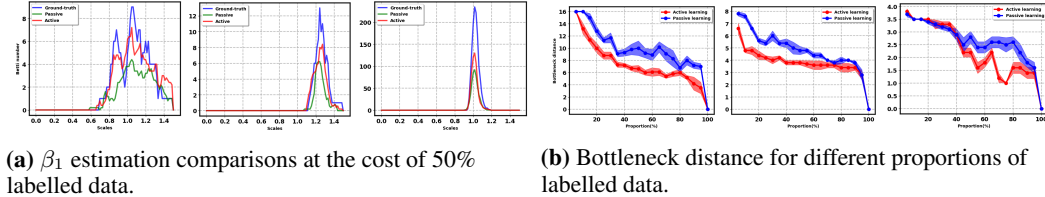

**(a)** $\beta_1$ estimation comparisons at the cost of 50% labelled data.

**(b)** Bottleneck distance for different proportions of labelled data.

**Figure 7:** Recovery of homological properties for the banknote (left), MNIST (middle) and CIFAR10 (right) using the active and passive learning approaches.

## 5 Conclusions

We propose an active learning algorithm to find the homology of decision boundaries. We theoretically analyze the query complexity of the proposed algorithm and prove the sufficient conditions to recover the homology of decision boundaries in the active learning setting. The extensive experiments on synthetic and real datasets with the application on model selection corroborate our theoretical results.

## Broader Impact

The proposed approach, although has strong algorithmic and theoretical merits, has potential real-world application as we demonstrated.

One of the key uses of this approach is to create efficient summaries of decision boundaries of datasets [23] and models. Such summaries can be quite useful in applications like AI model marketplaces [24], where data and models can be securely matched without revealing too much information about each other. This is helpful in scenarios where the data is private and models are proprietary or sensitive.

A downside of being able to compute homology of decision boundaries with few examples is that malicious users may be able to learn about the key geometric / topological properties of the models with fewer examples than they would use otherwise. While this in itself may be benign, combined with other methods, they may be able to design better adversarial attacks on this model for instance. Ways of mitigating it in sensitive scenarios include ensuring that users do not issue too many queries of examples close to the boundary successively, since this may be revealing of malicious intent.

## Acknowledgements

This work is funded in part by the Office of Naval Research under grant N00014-17-1-2826 and the National Science Foundation under grants OAC-1934766, CNS-2003081, and CCF-2007688.

## Footnotes

[1]Note that it is conceivable that the decision boundary is not strictly a manifold. While this assumption is critical to the rest of this paper, it is possible to extend the results here by following the theory in [12]. We will leave a thorough exploration of this for future work.

[2] A set $W$ is $\frac{\gamma}{2}$-dense in $\mathcal{M}$ if for every $\mathbf{p} \in \mathcal{M}$ there exists a $\mathbf{x} \in W$ such that $\|\mathbf{p} - \mathbf{x}\|_2 < \frac{\gamma}{2}$. In other words, there exists at least one $\mathbf{x} \in W$ in $B_{\frac{\gamma}{2}}(\mathbf{p})$ for every $\mathbf{p} \in \mathcal{M}$.

[3]The $k$-radius nearest neighbor graph connects all pairs of vertices that are a distance of at most $k$ away, and the $k$-nearest neighbor graph connects a vertex to its $k$ nearest neighbors

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
