[Supplementary Material]

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

# Appendices

In Appendix A, we present the sufficient conditions for estimating the homology of a manifold. In Appendix B, by assuming the decision boundaries is a manifold, we present the sample complexity result to generate a simplical complex homotopy equivalent to the decision boundary manifold. This is the passive learning result that we extend to the active learning case. In Appendix C, we present the shortest shortest ($S^2$) path algorithm [15]. As a graph-based active learning algorithm for nonparametric classification, $S^2$ is used in the label query of our proposed active learning framework (see Figure 2 in the main document). After these three preliminary sections, in Appendix D, we provide a complete proof of Theorem 1 in the main content. In Appendix E, we provide numerical comparisons of the query complexity between the passive learning and our active learning algorithm for finding the homology of decision boundaries. Lastly, we present our complete experimental results in Appendix F.

## A   Sufficient Conditions for Finding the Homology of a Manifold

In [13], the authors show how to *learn* the homology of a manifold from samples. Specifically, [13] assumes the samples are generated from a constrained domain, and with these generated samples, [13] provide the sufficient conditions to learn the manifold the samples lie on or nearby. The sufficient conditions are used to find the homology of decision boundaries in both passive learning setting [5] and active learning setting of our work. Hence we describe those conditions in this section. To begin with, several assumptions need to be made.

**Assumption A.1.** *The manifold $\mathcal{M}$ has a condition number $1/\tau$.*

The quantity $\tau$ associated with $\mathcal{M}$ encodes both the local and global curvature of the manifold and is linked with the intrinsic complexity of the manifold $\mathcal{M}$: If $\mathcal{M}$ consists of several components, then $\tau$ bounds the separation between them. For example, as shown in Figure A.1, if $\mathcal{M}$ is a sphere, then $\tau$ is the radius of the sphere.

In the setting of [13], one has access to points that are near $\mathcal{M}$. Let $\mathcal{D} = (\mathbf{x}_1, ..., \mathbf{x}_N)$ denote a set of these sample points drawn from a feature space/domain $\mathcal{X}$. Furthermore, we define $\mu_\mathcal{X}$ as a standard Lebesgue measure on $\mathcal{X}$ and we write $Tub_r(\mathcal{M})$ to denote a tubular neighborhood with radius $r$ around $\mathcal{M}$. As the goal is to learn $\mathcal{M}$ from samples, the domain $\mathcal{X}$ has to be *close* to $\mathcal{M}$ in some sense. To formalize this, the authors in [13] make the following assumption.

**Figure A.1:** An example of $(\epsilon, \gamma)-$labeled Čech complex, constructed in a tubular neighborhood $\mathrm{Tub}_{w+\gamma}(\mathcal{M})$ of radius $w + \gamma$, for a manifold $\mathcal{M}$ of condition number $1/\tau$. The overlap between the two classes is contained in $\mathrm{Tub}_w(\mathcal{M})$. The complex is constructed on samples in class 0, by placing balls of radius $\epsilon$ ($B_\epsilon(\mathbf{x}_i)$), and is "witnessed" by samples in class 1. $\mathcal{X}$ is the compact probability space for the data. Each triangle is assumed to be a $2-$simplex in the simplicial complex. Note that we keep the samples from both classes sparse for aesthetic reasons.

**Assumption A.2.** *The domain $\mathcal{X}$ is contained in a $\mathrm{Tub}_r(\mathcal{M})$ for $r < (\sqrt{9} - \sqrt{8})\tau$*

Assumption A.2 specifies that the generated samples are within at-most a distance of $r$ to $\mathcal{M}$. In addition, we also define a notion of density on $\mathcal{M}$.

**Definition A.1.** *The set $\mathcal{D}$ is said to be $r-$dense in $\mathcal{M}$ if for every $\mathbf{p} \in \mathcal{M}$ there exists some $\mathbf{x} \in \mathcal{D}$ such that $\|\mathbf{p} - \mathbf{x}\|_2 < r$.*

This ensures a sufficient mass of samples are generated near $\mathcal{M}$. Typically, for learning a manifold $\mathcal{M}$ and then finding the homology of $\mathcal{M}$, one need to construct $\epsilon$-balls centered at the points of $\mathcal{D}$ such that the union $U = \bigcup_{\mathbf{x} \in \mathcal{D}} B_\epsilon(\mathbf{x})$ deformation retracts to $\mathcal{M}$. [13] presents the sufficient conditions for having $U = \bigcup_{\mathbf{x} \in \mathcal{D}} B_\epsilon(\mathbf{x})$ be homotopy equivalent to $\mathcal{M}$:

**Theorem A.1.** *$\mathcal{M}$ is a deformation retract of $U = \bigcup_{\mathbf{x} \in \mathcal{D}} B_\epsilon(\mathbf{x})$ if **(a)** $r \leq (\sqrt{9} - \sqrt{8})\tau$, **(b)** $\epsilon \in \left( \frac{(r+\tau) - \sqrt{r^2 + \tau^2 - 6\tau r}}{2}, \frac{(r+\tau) + \sqrt{r^2 + \tau^2 + 6\tau r}}{2} \right)$, and **(c)** $\mathcal{D}$ is $r$-dense in $\mathcal{M}$.*

The proof of Theorem A.1 is under proposition 7.1 in [13]. Clearly, conditions (a) and (b) establish the relationship between $\epsilon$, $r$ and $\tau$. Condition (c) ensures there are sufficient samples covering $\mathcal{M}$.

**Remark A.1.** *Theorem A.1 states, with appropriately constructed $B_\epsilon(\mathbf{x})$, one can find a $U = \bigcup_{\mathbf{x} \in \mathcal{D}} B_\epsilon(\mathbf{x})$ where $\mathcal{M}$ is a deformation retract. This naturally turns directly accessing the homolicial features of $\mathcal{M}$ to finding the homology of the auxiliary $U$. A practical way to do the above is construct the simplicial complex of the cover of $U$ and then compute the homological properties from the constructed simplicial complex.*

One instructive way to think about the difference between the current paper and Niyogi *et al.*[13] is to think of the *sampling mechanism* of the data. The sampling mechanism in [13] essentially generates points on or very close to the manifold (depending on the amount of noise - see Assumption A.1). In our setting, however, we suppose that the feature data is supported on a much larger space $\mathcal{X}$, and there is no direct access to a manifold sampling mechanism. Instead, we have access to a *labeling oracle* which indirectly clues to us the location of the manifold. We therefore cannot make an assumption as strong as Assumption A.1, and we will relax this in Appendix B.

# B    The Manifold of Decision Boundaries

A recent line of work [9, 4] has emerged to find the homology of decision boundaries. The decision boundaries are simply taken as a manifold, and in this way, one can capture the holomoloical features of the decision boundaries by finding the homology of the related manifold. In the sequel, by referring to [5], we first describe the setting of a classification problem, then present a special simplicial complex called labeled Čech (LČ) complex, and finally present the sample complexity result in the passive learning setting through the LČ complex.

We consider a binary classification problem such that $\mathcal{D}$ and labels $y \in \{0, 1\}$ are drawn from joint distribution $p_{XY}$. For the distribution $p_{XY}$, we will let

$$\mathfrak{D} = \{\mathbf{x} \in \mathcal{X} : p_{X|Y}(\mathbf{x} \mid 1)p_{X|Y}(\mathbf{x} \mid 0) > 0\}.$$

In other words, $\mathfrak{D}$ denotes the region of the feature space where both classes overlap, i.e., both class conditional distributions $p_{X|Y}(\cdot \mid 1)$ and $p_{X|Y}(\cdot \mid 0)$ have non-zero mass. Similar to the notation $\mu_{\mathcal{X}}$, we write $\mu_{\mathcal{X}|y}$ to denote the measure for class $y$ on $\mathcal{X}$. **From here on, we reuse the notation $\mathcal{M}$ to denote the manifold of the decision boundaries.** Specially, we define $\mathcal{M} = \{\mathbf{x} \in \mathcal{X}|p_{Y|X}(1|\mathbf{x}) = p_{Y|X}(0|\mathbf{x})\}$. The optimal decision boundaries is given by the classification function

$$f(\mathbf{x}) = \begin{cases} 1 & \text{if } p_{Y|X}(1 \mid \mathbf{x}) \geq 0.5 \\ 0 & \text{otherwise} \end{cases}.$$

Based on the observed data, we define the set $\mathcal{D}^0 = \{\mathbf{x} \in \mathcal{D} : f(\mathbf{x}) = 0\}$, that is the set of all samples with a Bayes optimal label of 0; similarly, we let $\mathcal{D}^1 = \{\mathbf{x} \in \mathcal{D} : f(\mathbf{x}) = 1\}$. Furthermore, we write $\text{Tub}_w(\mathcal{M})$ to denote the smallest tubular neighborhood enclosing $\mathfrak{D}$.

**Definition B.1.** *Given $\epsilon, \gamma > 0$, an $(\epsilon, \gamma)$-labeled Čech complex is a simplicial complex constructed from a collection of simplices such that each simplex $\sigma$ is formed on the points in a set $S \subseteq \mathcal{D}^0$ witnessed by the reference set $\mathcal{D}^1$ satisfying the following conditions: (a) $\bigcap_{\mathbf{x}_i \in \sigma} B_\epsilon(\mathbf{x}_i) \neq \emptyset$, where $\mathbf{x}_i \in S$ are the vertices of $\sigma$. (b) $\forall \mathbf{x}_i \in S \subseteq \mathcal{D}^0$, $\exists \mathbf{x}_j \in \mathcal{D}^1$ such that, $\|\mathbf{x}_i - \mathbf{x}_j\|_2 \leq \gamma$.*

Definition B.1(a) ensures the LČ complex is constructed following the definitions of a typical Čech complex, that is, a set of $\epsilon$ ball centered at points of $\sigma$ has a non-empty intersection [25]. Definition B.1(b) states that a subset $S$ of $\mathcal{D}^0$ is selected such that $S$ is at-most $\gamma$ distance to $\mathcal{D}^1$. The authors in [5] show that, under certain assumption on the manifold and the distribution, provided that sufficiently many random samples (and their labels) are drawn according $p_{XY}$ then $U = \bigcup_{\mathbf{x}_i \in \sigma} B_\epsilon(\mathbf{x}_i)$ is homotopy equivalent to $\mathcal{M}$. Therefore one could estimate the homologial features of the decision boundaries through the LČ complex. Herein, we introduce the assumptions and the lemmas before getting to the sample complexity result for the passive learning. These assumptions are standard in deriving theoretical results with respect to a manifold as explained in [13]. Hence they are also critical to the main theoretical results of this paper.

Note that the principal difference between [13] and our work is that [13] assumes all generated samples are contained within $\text{Tub}_{(\sqrt{9}-\sqrt{8})\tau}(\mathcal{M})$ (See Assumption A.2); this is one of the sufficient conditions for manifold reconstruction from samples. In contrast, [5] considers a more practical scenario for classification where samples can be generated anywhere following the joint distribution $p_{XY}$. Our approach more closely follows [5], but with an important distinction. We sample from the marginal distribution $p_X$ without knowledge of the underlying labels and use active learning to discover the labels. Nevertheless, our work and [5] share the assumptions and the lemmas introduced in the sequel. We also emphasize that, as we would show in the end, the follow-on results are dependent on two parameters that are specific to the joint distribution: $w$ - the amount of overlap between the distributions and $\tau$ - the global geometric property of the decision boundary manifold.

**Assumption B.1.** $w < (\sqrt{9} - \sqrt{8})\tau$.

Since the generated samples do not necessarily reside within $\text{Tub}_{(\sqrt{9}-\sqrt{8})\tau}(\mathcal{M})$, it is not immediately apparent that it is possible to find an $S$ (see Definition B.1) that is entirely contained in $\text{Tub}_{(\sqrt{9}-\sqrt{8})\tau}(\mathcal{M})$. However, Assumption B.1 allows us to guarantee precisely this. To see this, we will first state the following lemma.

**Lemma B.1.** *Provided $\mathcal{D}^0$ and $\mathcal{D}^1$ are both $\frac{\gamma}{2}$-dense in $\mathcal{M}$, then $S$ is contained in $Tub_{w+\gamma}(\mathcal{M})$ and it is $\frac{\gamma}{2}$-dense in $\mathcal{M}$.*

Assumption B.1 is imposed on $p_{XY}$ therefore there exists a $S$ contained in $\text{Tub}_{(\sqrt{9}-\sqrt{8})\tau}(\mathcal{M})$ with a proper $\gamma$. In fact, $S$ can be partitioned to $S_{\mathfrak{D}} = S \bigcap \mathfrak{D}$ and $S \backslash S_{\mathfrak{D}}$. Under Assumption B.1, we immediately have $S_{\mathfrak{D}} \subseteq \text{Tub}_w(\mathcal{M}) \subset \text{Tub}_{(\sqrt{9}-\sqrt{8})\tau}(\mathcal{M})$. $S \backslash S_{\mathfrak{D}}$ is actually an excess from $\mathfrak{D}$ caused by identifying the necessary points $\mathbf{x} \in \mathcal{D}^0$ in the construction of LČ complex (see (b) in Definition B.1), and the exceeding extent is controlled by $\gamma$. With $\gamma < (\sqrt{9} - \sqrt{8})\tau - w$, we immediately have $S \subset \text{Tub}_{(\sqrt{9}-\sqrt{8})\tau}(\mathcal{M})$. Besides, as the distance from $\mathbf{x}_i \in S \subseteq \mathfrak{D}^0$ to $\mathbf{x}_j \in \mathcal{D}^1$ is bounded by $\gamma$ (see definition B.1(b)), $S$ is also $\frac{\gamma}{2}$-dense in $\mathcal{M}$ therefore implying $(w + \gamma)$-dense in $\mathcal{M}$.

In fact, having $S \subset \text{Tub}_{(\sqrt{9}-\sqrt{8})\tau}(\mathcal{M})$, $S$ being $(w + \gamma)$-dense in $\mathcal{M}$ and $\epsilon$ properly selected are the sufficient conditions (see Theorem A.1) to construct a $U = \bigcup_{\mathbf{x}_i \in \sigma} B_\epsilon(\mathbf{x}_i)$ homotopy equivalent to $\mathcal{M}$. Remembering that the LČ complex is the cover of $U$, we have the following proposition

**Proposition B.1.** *$(\epsilon, \gamma)$-LČ complex is homotopy equivalent to $\mathcal{M}$ as long as (a) $\gamma < (\sqrt{9} - \sqrt{8})\tau - w$; (b) $\mathcal{D}^0$ and $\mathcal{D}^1$ are $\frac{\gamma}{2}$-dense in $\mathcal{M}$ and (c) $\epsilon \in (\frac{(w+\gamma+\tau)-\sqrt{(w+\gamma)^2+\tau^2-6\tau(w+\gamma)}}{2}, \frac{(w+\gamma+\tau)+\sqrt{(w+\gamma)^2+\tau^2-6\tau(w+\gamma)}}{2})$.*

A pictorial description of relations between $w$, $\gamma$ and $\tau$ is in Figure A.1 in the main content. In the stylized example in Figure A.1, $\mathcal{M}$ is a circle, $\text{Tub}_{w+\gamma}(\mathcal{M})$ is an annulus and the radius $\epsilon$ of the covering ball $B_\epsilon(\mathbf{x})$ is constrained by $\tau$.

In [5], the authors derive a sample complexity result in a passive learning setting such that a LČ complex is homotopy equivalent to $\mathcal{M}$. The heart of the derivation is figuring out how many samples are needed to have $\mathcal{D}^0$ and $\mathcal{D}^1$ both $\frac{\gamma}{2}$-dense in $\mathcal{M}$. We require an additional assumption:

**Assumption B.2.** $\inf_{\mathbf{x} \in \mathcal{M}} \mu_{\mathcal{X}|y}(B_{\gamma/4}(\mathbf{x})) > \rho_{\gamma/4}^y, y \in \{0, 1\}$.

The Assumption B.2 ensures sufficient mass in both classes such that $\mathcal{D}^0$ and $\mathcal{D}^1$ are $\frac{\gamma}{2}$-dense in $\mathcal{M}$. Given the Proposition B.1, this leads to the following theorem presented in Theorem 3 in [5].

**Theorem B.1.** *Let $N_{\gamma/4}$ be the covering number of the manifold $\mathcal{M}$. Under Assumptions A.1 B.1 and B.2, for any $\delta > 0$, we have that the $(\epsilon, \gamma)$-LČ complex constructed from $\mathcal{D}_0$ and $\mathcal{D}_1$ is homotopy equivalent to $\mathcal{M}$ with probability at least $1 - \delta$ provided*

$$|\mathcal{D}| > \max \left\{ \frac{1}{P(y=0)\rho_{\gamma/4}^0} \left[ \log\left(2N_{\gamma/4}\right) + \log\left(\frac{1}{(\delta)}\right) \right], \frac{1}{P(y=1)\rho_{\gamma/4}^1} \left[ \log\left(2N_{\gamma/4}\right) + \log\left(\frac{1}{(\delta)}\right) \right] \right\}$$
(B.1)

The complete proof is elaborated in [5].

**Remark B.1.** *Considering the selection of $\gamma$ needs to follow the Proposition B.1(b) as a function of $\tau$ and $w$, therefore given a probability factor $\delta$, the sample complexity is dependent on $w$ and $\tau$.*

## C  Shortest Shortest Path Algorithm

We use an algorithm called shortest shortest $(S^2)$ path [15] to query labels in the proposed framework. As a graph-based active learning algorithm for binary classification, $S^2$ has a properties of efficiently revealing the vertices near the cut-edges of a graph. Therefore, we apply $S^2$ to the label query stage of our proposed active learning framework (See Figure 2). The details of $S^2$ are presented in Algorithm 1. Repetitively using the defined notations, we let $G = (\mathcal{D}, E)$ denote a graph constructed from $\mathcal{D}$. This allows us to define the cut-set $C = \{(\mathbf{x}_i, \mathbf{x}_j) | y_i \neq y_j \wedge (\mathbf{x}_i, \mathbf{x}_j) \in E\}$ and cut-boundary $\partial C = \{\mathbf{x} \in \mathcal{D} : \exists e \in C \text{ with } \mathbf{x} \in e\}$. $S^2$ functions to efficiently identify $\partial C$ by querying labels based on the structure of $G$. The query process is split to a uniform sampling phase and a path bisection phase. The uniform sampling serves to find a path connecting vertices of opposite labels. This corresponds to line 3 in Algorithm 1. The path bisection phase queries the mid-point of the shortest path that connects oppositely labeled vertices in the underlying graph. This corresponds to line 10 in Algorithm 1. The BUDGET in Algorithm 1 represents the cardinality of the query set $\tilde{\mathcal{D}}$. $S^2$ is designed in a consideration of running limited expensive number of label queries on vertices, as

| **Algorithm 1:** $S^2$: Shortest Shortest Path | **Procedure 1:** MSSP: mid-point of the shortest shortest path |
|---|---|
| **Input**: Graph $G = (\mathcal{D}, E)$, BUDGET$\leq N$ | **Input**: $G = (\mathcal{D}, E), \tilde{\mathcal{D}} \subseteq \mathcal{D}$ |
| 1: $\tilde{\mathcal{D}} \leftarrow \emptyset$ | 1: **for** each $\mathbf{x}_i, \mathbf{x}_j \in \tilde{\mathcal{D}}$ such that $f(\mathbf{x}_i) \neq f(\mathbf{x}_j)$ |
| 2: **while** 1 **do** | 2: $\quad P_{ij} \leftarrow$ shortest path between $\mathbf{x}_i$ and $\mathbf{x}_j$ in $G$ |
| 3: $\quad$ $\mathbf{x} \leftarrow$ Randomly chosen unlabeled vertex | 3: $\quad \mathcal{L}_{ij} \leftarrow$ length of $P_{ij}$ ($\infty$ if no path exists) |
| 4: $\quad$ **do** | 4: **end for** |
| 5: $\quad\quad$ Add $(\mathbf{x}, f(\mathbf{x}))$ to $\tilde{\mathcal{D}}$ | 5: $i^*, j^* \leftarrow argmin_{\mathbf{x}_i, \mathbf{x}_j \in \tilde{\mathcal{D}}: f(\mathbf{x}_i) \neq f(\mathbf{x}_j)} \mathcal{L}_{ij}$ |
| 6: $\quad\quad$ Remove all the found cut-edges of G | 6: **if** $(i^*, j^*)$ exists **then** |
| 7: $\quad\quad$ **if** $|\tilde{\mathcal{D}}|$ = BUDGET **then** | 7: $\quad$ **Return** mid-point of $P_{i^* j^*}$ |
| 8: $\quad\quad\quad$ **Return** $\tilde{\mathcal{D}}$ | 8: **else** |
| 9: $\quad\quad$ **end if** | 9: $\quad$ **Return** $\quad \emptyset$ |
| 10: $\quad$ **while** $x \leftarrow$ MSSP$(G, \tilde{\mathcal{D}})$ exists | 10: **end if** |
| 11: **end while** | |

a result, efficiently finding the vertices of cut-edges with the limited budget. Based on Theorem 1 in [15], we provide a simplified query complexity result for recovering $\partial C$:

**Theorem C.1.** *Suppose a graph $G = (\mathcal{D}, E)$ with a binary function $f : \mathcal{D} \to \{0, 1\}$ partitioning the graph $G$ into two components identically labeled. Let $\beta$ denote the proportion of the smallest components. Then for any $\delta > 0$, $S^2$ will recover $C$ with probability at least $1 - \delta$ if the complexity of queries is at least*

$$\frac{log(1/(\beta\delta))}{log(1/(1-\beta))} + |\partial C|(\lceil log_2|\mathcal{D}|\rceil + 1) \tag{C.1}$$

The query complexity is simplified by upper-bounding the path length with $|\mathcal{D}|$ and the complete proof can be found in [15].

# D  The Query Complexity Proof for the Proposed Active Learning Algorithm

The proof of Theorem 1 in our main content comprises of upper-bounding the query complexity to identify our target $\partial C$ and upper-bounding the sample complexity of having $\mathcal{D}^0$ and $\mathcal{D}^1$ both $\frac{\gamma}{2}$-dense in $\mathcal{M}$. **We recap Theorem 1 in the main content as the Theorem D.1 in as follows and provide the complete proof.** Before we move to the proof, we make one last assumption,

**Assumption D.1.** *(a)* $\sup_{\mathbf{x} \in \mathcal{M}} \mu_{\mathcal{X}}(B_{(w+\gamma)}(\mathbf{x})) < h_{(w+\gamma)}$. *(b)* $\mu_{\mathcal{X}}(\text{Tub}_{w+\gamma}(\mathcal{M})) \leq N_{w+\gamma} h_{w+\gamma}$.

Assumption D.1 upper-bounds the measure of $\text{Tub}_{w+\gamma}(\mathcal{M})$. Besides, for the convenience of the theorem derivation, we use $k$-radius neighbor paradigm to construct $G = (D, E)$. We also need the following lemma,

**Lemma D.1.** *Suppose $\mathcal{D}^0$ and $\mathcal{D}^1$ are $\frac{\gamma}{2}$-dense in $\mathcal{M}$, then the graph $G = (\mathcal{D}, E)$ constructed from $\mathcal{D}$ is such that $\mathcal{D}^0 \bigcap \partial C$ and $\mathcal{D}^1 \bigcap \partial C$ are both $\frac{\gamma}{2}$-dense in $\mathcal{M}$ and $\partial C \subseteq \text{Tub}_{w+\gamma}(\mathcal{M})$ for $k = \gamma$.*

$\mathcal{D}^0$ and $\mathcal{D}^1$ being $\frac{\gamma}{2}$-dense in $\mathcal{M}$ indicates the longest distance between $\mathbf{x}_i \in \mathcal{D}^0 \bigcap B_{\frac{\gamma}{2}}(\mathbf{p})$ and $\mathbf{x}_j \in \mathcal{D}^1 \bigcap B_{\frac{\gamma}{2}}(\mathbf{p})$ for $\mathbf{p} \in \mathcal{M}$ is $\gamma$. Therefore, letting $k = \gamma$ as Lemma D.1 suggests will result in $\mathcal{D}^0 \bigcap \partial C$ and $\mathcal{D}^1 \bigcap \partial C$ both being $\frac{\gamma}{2}$-dense in $\mathcal{M}$. Similar to Lemma B.1, constructing a graph with a $\gamma$ radius inevitably results in a subset of points of $\partial C$ leaking out of $\text{Tub}_w(\mathcal{M})$ and we formally have $\partial C \subseteq \text{Tub}_{w+\gamma}(\mathcal{M})$. Proposition B.1 states the proper choice of $\gamma$. With the above introduced, one can see the key intuition behind our approach is turning $S^2$ to focus on labels of points falling within $\text{Tub}_{w+\gamma}(\mathcal{M})$. As we show below, this is done in a remarkably query efficient manner; when the labeled data is obtained we can construct an LČ complex and this allows us to find the homology of the manifold $\mathcal{M}$.

**Theorem D.1.** *Let $N_{w+\gamma}$ be the covering number of the manifold $\mathcal{M}$. Under Assumptions A.1 B.1 B.2 and D.1, for any $\delta > 0$, we have that the $(\epsilon, \gamma)$-LČ complex estimated by our framework is homotopy equivalent to $\mathcal{M}$ with probability at least $1 - \delta$ provided*

$$|\tilde{\mathcal{D}}| > \frac{\log\left\{1/\left[\beta\left(1 - \sqrt{1 - \delta}\right)\right]\right\}}{\log\left[1/(1 - \beta)\right]} + |\mathcal{D}|N_{w+\gamma}h_{w+\gamma}(\lceil\log_2|\mathcal{D}|\rceil + 1) \tag{D.1}$$

*where*

$$|\mathcal{D}| > \max\left\{\frac{1}{P(y=0)\rho^0_{\gamma/4}}\left[\log\left(2N_{\gamma/4}\right) + \log\left(\frac{1}{(1 - \sqrt{1 - \delta})}\right)\right],\right. \\ \left.\frac{1}{P(y=1)\rho^1_{\gamma/4}}\left[\log\left(2N_{\gamma/4}\right) + \log\left(\frac{1}{(1 - \sqrt{1 - \delta})}\right)\right]\right\} \tag{D.2}$$

*Proof.* Let $E_a$ denote an event that $\mathcal{D}^0$ and $\mathcal{D}^1$ are both $\frac{\gamma}{2}$-dense in $\mathcal{M}$. Let $E_b$ denote an event that the LČ complex constructed from the query set $\tilde{\mathcal{D}}$ is homotopy equivalent to $\mathcal{M}$. Clearly $E_b$ never happens if $E_a$ does not happen due to not satisfying the condition (b) in Proposition B.1, and this results the conditional probability $P(E_b|\overline{E_a}) = 0$. Now, we expand the probability of $E_b$ as follow:

$$P(E_b) = P(E_b|E_a)P(E_a) + P(E_b|\overline{E_a})P(\overline{E_a})$$
$$= P(E_b|E_a)P(E_a) \tag{D.3}$$

We first prove a query complexity result for the event $E_b|E_a$, i.e., how likely the event $E_b|E_a$ would happen with a certain amount of queries. Similarly, Eq. B.1 already provides a sample complexity result on the occurrence of the event $E_a$. With the Eq. D.3, we unify both complexity results and derive the theorem. We now consider $P(E_b|E_a)$. The query set $\tilde{\mathcal{D}}$ requires that $\tilde{\mathcal{D}}^0$ and $\tilde{\mathcal{D}}^1$ are $\frac{\gamma}{2}$-dense in $\mathcal{M}$ for $E_b$ to happen. This can be surely achieved by constructing appropriate $k$-radius near neighbor/$k$ nearest neighbor graph $G = (\mathcal{D}, E)$ stated by Lemma D.1, provided $\partial C \subseteq \tilde{\mathcal{D}}$. Hypothetically if a proper $k$ for construction of $G$ is selected, the event $E_b$ becomes completely identifying the examples in $\partial C$ through label querying. Theorem C.1 upper-bounds the query complexity of finding $\partial C$ with a probably correct result. As the Assumption D.1 holds, we further upper-bound $|\partial C|$ by $|\partial C| \leq |\mathcal{D}|N_{w+\gamma}h_{w+\gamma}$. This gives us the query complexity of having $E_b|E_a$ happen:

$$|\tilde{\mathcal{D}}| > \frac{log(1/(\beta\alpha))}{log(1/(1 - \beta))} + |\mathcal{D}|N_{w+\gamma}h_{w+\gamma}|log_2|\mathcal{D}| \tag{D.4}$$

with the probability at least $1 - \alpha$. We now turn to $P(E_a)$. The occurrence of $E_a$ is a dual implication of the LČ complex constructed from $\mathcal{D}$ being homotopy to $\mathcal{M}$. Therefore, reusing the result in Theorem B.1 (Eq. B.1), we get if

$$|\mathcal{D}| > max\left(\frac{1}{P(y=0)\rho^0_{\gamma/4}}\left(log\left(2N_{\gamma/4}\right) + log\left(\frac{1}{\eta}\right)\right), \frac{1}{P(y=1)\rho^1_{\gamma/4}}\left(log\left(2N_{\gamma/4}\right) + log\left(\frac{1}{\eta}\right)\right)\right) \tag{D.5}$$

then $E_a$ happens with the probability at least $1 - \eta$.

Picking $1 - \alpha = \sqrt{1 - \delta}$ and $1 - \eta = \sqrt{1 - \delta}$, we can unify the sample complexity results of $E_b|E_a$ and $E_a$ to $E_b$ and complete the proof. $\square$

# E   Numerical Comparison for the Active Learning and Passive Learning Algorithms

As Eq. D.1 and Eq. B.1 directly provide the upper-bound query/sample complexity results of the active learning and passive learning methods, we can numerically compare the two methods. Herein, we provide a description of the evaluation; for implementation details we refer the reader to our code.

We created a stylized example illustrated in Figure A.1. We assume the feature space/domain $\mathcal{X}$ is a square area. In the domain $\mathcal{X}$, we draw samples generated from $p_{XY}$ with a circular decision boundary of radius $\tau$. We further use $w$ to denote the radius of a smallest $\mathrm{Tub}_r(\mathcal{M})$ to enclose the overlap $\mathfrak{D}$ between two classes. Both $\tau$ and $w$ are intrinsic properties of $\mathcal{M}$ and $p_{XY}$. Given these two properties, we set $\gamma = (\sqrt{9} - \sqrt{8})\tau - w - 10^{-5}$ to satisfy Proposition B.1(b). We make several additional assumptions regarding the problem in order to conduct the numerical experiments. Let us suppose that $\mathcal{X}$ is a square of $5 \times 5$ units. We assume $\mathcal{D}^0$ and $\mathcal{D}^1$ are uniformly distributed in the subspace $\mathcal{X}_0 \subseteq \mathcal{X}$ and subspace $\mathcal{X}_1 \subseteq \mathcal{X}$. Let $\mathcal{X}_0 \bigcap \mathcal{X}_1 = \mathrm{Tub}_w(\mathcal{M})$ and $\mathcal{X}_0 \bigcup \mathcal{X}_1 = \mathcal{X}$. Class-conditional distributions $p_{X|0}$ and $p_{X|1}$ are both uniform density functions in $\mathcal{X}_0$ and $\mathcal{X}_1$ such that class 0 and 1 completely overlap in $\mathrm{Tub}_w(\mathcal{M})$. Furthermore, we have $Area(\mathcal{X}_0) = Area(\mathcal{X}) - \pi(\tau - w)^2 = 25 - \pi(\tau - w)^2$ and $Area(\mathcal{X}_1) = \pi(\tau + w)^2$. Having the uniform probability density $d_0 = \frac{1}{Area(\mathcal{X}_0)}$ for class 0 and $d_1 = \frac{1}{Area(\mathcal{X}_1)}$ for class 1, we can easily compute the actual values of $h_{(w+\gamma)} = \mu_{\mathcal{X}}(B_{w+\gamma}(\mathbf{x}))$, $\rho_{\gamma/4}^0 = \mu_{\mathcal{X}|0}(B_{\gamma/4}(\mathbf{x}))$ and $\rho_{\gamma/4}^1 = \mu_{\mathcal{X}|1}(B_{\gamma/4}(\mathbf{x}))$ in Eq. B.1 and Eq. D.1 by simple algebra operations. $N_{\gamma/4}$ in Eq. B.1 indicates the cover number of $\mathcal{M}$ realized by $\frac{\gamma}{4}-$ balls. We simulate $N_{\gamma/4}$ by covering $\mathcal{M}$ with least number of $B_{\gamma/4}(\mathbf{x})$ on $\mathcal{M}$. The same operations can be applied to obtain $N_{w+\gamma}$ in Eq. D.1. $\beta$ in Eq. D.1 indicates the proportion of the smallest component with the datapoints identically labelled in $G = (\mathcal{D}, E)$. For $G$ constructed by the datapoints in our created stylized example, there are only two such components thus each component contains all the datapoints from class 0 or 1. Therefore, $\beta$ is same as the mixture probability where $\beta = P(y = 1)$. We set $P(y = 1) = \frac{\pi\tau^2}{25}$ such that the probability accessing $\mathcal{M}$ by samples generated from $p_{XY}$ increases with $\tau$.

We compare the sample complexity results by fixing $w$ and varying $\tau$ or fixing $\tau$ and varying $w$. For the case of fixing $w$, we vary $\tau$ from 0.1 to 0.7 and set $\delta = 0.1$ and $w = 10^{-10}$. For the case of fixing $\tau$, on the other hand, we vary $w$ from $10^{-10}$ to $1.75 \times 10^{-2}$ and fix $\delta = 0.1$ and $\tau = 0.1$. Having $w$, $\tau$ and $\delta$, we quantify other variables in Eq. D.1 and Eq. B.1 with the method described above and therefore acquire the query complexity for the active learning and the sample complexity for the passive learning. We calculate the ratio of the query complexity to the sample complexity and the results are shown in Figure E.1. As expected, the proposed active learning algorithm has a significant complexity gain compared to the passive learning case, especially for smaller values of $\tau$ and $w$.

**(a)** Varying $\tau$.      **(b)** Varying $w$.

**Figure E.1:** The ratio of query complexity to sample complexity by varying $\tau$ or $w$.

# F    Complete Experimental Results

We provide comprehensive performance results evaluated from using the characteristics of homology group of dimension 0 ($\beta_0$, PD$_1$) and dimension 1 ($\beta_1$, PD$_1$).

## F.1    Experimental Results on Synthetic Data

**Figure F.1:** $\beta_0$ estimated at the cost of different proportion of unlabelled data pool by the passive learning (first row) and active leanring methods with $0.25$ (second), $0.45$ (third) and $0.65$ (forth) radius near neighbors graphs.

**Figure F.2:** $\beta_1$ estimated at the cost of different proportion of unlabelled data pool by the passive learning (first row) and active leanring methods with $0.25$ (second), $0.45$ (third) and $0.65$ (forth) radius near neighbors graphs.

**(a)** Bottleneck distance from the ground-truth $PD_0$     **(b)** Bottleneck distance from the ground-truth $PD_1$

**Figure F.3:** Bottleneck distance from the ground-truths $PD_0$ and $PD_1$ by the passive learning and active learning on synthetic data. $k$ indicates the values used in the $k$-radius near neighbor graphs for the proposed active learning algorithm.

## F.2  Experimental Results on Real Data

**(a)** Bottleneck distance from the ground-truth $PD_0$

**(b)** Bottleneck distance from the ground-truth $PD_1$

**Figure F.4:** Bottleneck distance from the ground-truths $PD_0$ and $PD_1$ by the passive learning and active learning on **Banknote**. $k$ indicates the values used in the $k$-nearest neighbor graphs for the proposed active learning algorithm.

**(a)** Bottleneck distance from the ground-truth $PD_0$

**(b)** Bottleneck distance from the ground-truth $PD_1$

**Figure F.5:** Bottleneck distance from the ground-truths $PD_0$ and $PD_1$ by the passive learning and active learning on **MNIST**. $k$ indicates the values used in the $k$-nearest neighbor graphs for the proposed active learning algorithm.

**(a)** Bottleneck distance from the ground-truth $PD_0$

**(b)** Bottleneck distance from the ground-truth $PD_1$

**Figure F.6:** Bottleneck distance from the ground-truths $PD_0$ and $PD_1$ by the passive learning and active learning on **CIFAR10**. $k$ indicates the values used in the $k$-nearest neighbor graphs for the proposed active learning algorithm.

| Banknote | KNN | SVM | Neural network | Decision tree |
|---|---|---|---|---|
| Passive | 0.0508±0.0018 | **0.1673±0.0077** | 0.0223±0.0000 | 0.1491±0.0038 |
| Active$^1$ | 0.0635±0.0012 | 0.3031±0.0011 | **0.0157±0.0000** | 0.1613±0.0004 |
| Active$^2$ | 0.0689±0.0014 | **0.1112±0.0109** | 0.0676±0.0033 | **0.0984±0.0026** |
| Active$^3$ | **0.0193±0.0000** | 0.1673±0.0077 | 0.0464±0.0021 | 0.1648±0.0024 |
| Passive (ens) | 0.0176 ±0.0000 | **0.0259±0.0000** | 0.0068 0.0000 | 0.0736±0.0000 |
| Active$^1$ (ens) | 0.0173±0.0000 | **0.0259±0.0000** | **0.0039±0.0000** | **0.0731±0.0000** |
| Active$^2$ (ens) | **0.0149±0.0000** | **0.0259±0.0000** | 0.0134±0.0001 | **0.0731±0.0000** |
| Active$^3$ (ens) | **0.0149±0.0000** | **0.0259±0.0000** | 0.0072±0.0000 | 0.0770±0.0000 |
| **MNIST** | KNN | SVM | Neural network | Decision tree |
| Passive | 0.0150±0.0000 | 0.0303±0.0000 | 0.1053±0.0000 | 0.0420±0.0003 |
| Active$^1$ | **0.0124±0.0000** | **0.0255±0.0001** | **0.0302±0.0004** | **0.0382±0.0001** |
| Active$^2$ | 0.0144±0.0000 | 0.0272±0.0001 | 0.0428±0.0012 | 0.0444±0.0002 |
| Active$^3$ | 0.0138±0.0000 | 0.0303±0.0001 | 0.0506±0.0010 | 0.0448±0.0002 |
| Passive (ens) | 0.0127 ±0.0000 | 0.0124±0.0000 | 0.0137 0.0000 | **0.0274±0.0000** |
| Active$^1$ (ens) | **0.0106±0.0000** | **0.0119±0.0000** | **0.0116±0.0000** | 0.0284±0.0000 |
| Active$^2$ (ens) | 0.0121±0.0000 | **0.0119±0.0000** | 0.0138±0.0000 | 0.0284±0.0000 |
| Active$^3$ (ens) | 0.0110±0.0000 | **0.0119±0.0000** | 0.0138±0.0000 | **0.0274±0.0000** |
| **CIFAR10** | KNN | SVM | Neural network | Decision tree |
| Passive | 0.3049±0.0004 | 0.4309±0.0010 | **0.2796±0.0009** | **0.3120±0.0000** |
| Active$^1$ | 0.3072±0.0004 | **0.4055±0.0000** | 0.2872±0.0006 | **0.3120±0.0000** |
| Active$^2$ | **0.2813±0.0000** | 0.4182±0.0006 | 0.3042±0.0006 | **0.3120±0.0000** |
| Active$^3$ | **0.2813±0.0000** | 0.4594±0.0008 | 0.3042±0.0006 | **0.3120±0.0000** |
| Passive (ens) | 0.2941±0.0002 | **0.2698±0.0000** | 0.2590 0.0001 | **0.3074±0.0000** |
| Active$^1$ (ens) | 0.2832±0.0000 | 0.2797±0.0000 | **0.2558±0.0000** | **0.3074±0.0000** |
| Active$^2$ (ens) | **0.2813±0.0000** | 0.2797±0.0000 | **0.2558±0.0000** | 0.3120±0.0000 |
| Active$^3$ (ens) | **0.2813±0.0000** | 0.2864±0.0003 | 0.2649±0.0001 | 0.3097±0.0000 |

**Table 2:** Average test error rates(five trials) on Banknote, MNIST and CIFAR10 for the model selected with 15% unlabelled pool data. Passive/Active stands for the non-ensemble classifiers selected by the $\mathbf{PD}_0$ homological similarities. Passive/Active (ens) stands for the classifiers ensembled from two classifiers: one is selected by the $\mathbf{PD}_0$ homological similarities and the other one is selected by the validation error. The subscript 1, 2 and 3 of the active learning indicates the used 3NN, 5NN and 7NN graphs. Best performance in the non-ensemble and ensemble cases are boldfaced.

| Banknote | KNN | SVM | Neural network | Decision tree |
|---|---|---|---|---|
| Passive | 0.1072±0.0000 | 0.3753±0.0005 | 0.4316±0.0000 | 0.1997±0.0000 |
| Active$^1$ | 0.0783±0.0014 | 0.3231±0.0012 | 0.4316±0.0000 | 0.1901±0.0004 |
| Active$^2$ | 0.1017±0.0001 | 0.3431±0.0012 | 0.3730±0.0138 | 0.1744±0.0026 |
| Active$^3$ | **0.0346±0.0013** | **0.0836±0.0133** | **0.1058±0.0265** | **0.1613±0.0004** |
| Passive (ens) | 0.0176±0.0000 | **0.0259±0.0000** | 0.0068±0.0000 | 0.0741±0.0000 |
| Active$^1$ (ens) | 0.0173±0.0000 | **0.0259±0.0000** | **0.0039±0.0000** | **0.0731±0.0000** |
| Active$^2$ (ens) | **0.0149±0.0000** | **0.0259±0.0000** | 0.0134±0.0001 | **0.0731±0.0000** |
| Active$^3$ (ens) | **0.0149±0.0000** | **0.0259±0.0000** | 0.0072±0.0000 | 0.0770±0.0000 |
| **MNIST** | KNN | SVM | Neural network | Decision tree |
| Passive | 0.0129±0.0000 | **0.0141±0.0000** | 0.0202±0.0000 | **0.0332±0.0000** |
| Active$^1$ | 0.0128±0.0000 | 0.0161±0.0001 | **0.0150±0.0000** | 0.0388±0.0001 |
| Active$^2$ | 0.0122±0.0000 | 0.0162±0.0001 | 0.0177±0.0000 | **0.0332±0.0000** |
| Active$^3$ | **0.0104±0.0000** | 0.0156±0.0001 | 0.0388±0.0020 | **0.0332±0.0000** |
| Passive (ens) | 0.0119 ±0.0000 | 0.0124±0.0000 | **0.0104±0.0000** | 0.0290±0.0000 |
| Active$^1$ (ens) | 0.0123±0.0000 | **0.0119±0.0000** | **0.0104±0.0000** | 0.0284±0.0000 |
| Active$^2$ (ens) | 0.0108±0.0000 | **0.0119±0.0000** | 0.0125±0.0000 | 0.0284±0.0000 |
| Active$^3$ (ens) | **0.0104±0.0000** | **0.0119±0.0000** | 0.0127±0.0000 | **0.0274±0.0000** |
| **CIFAR10** | KNN | SVM | Neural network | Decision tree |
| Passive | **0.3065±0.0002** | 0.4683±0.0000 | 0.3185±0.0000 | **0.3625±0.0000** |
| Active$^1$ | 0.3201±0.0000 | 0.4591±0.0005 | **0.3058±0.0006** | **0.3625±0.0000** |
| Active$^2$ | 0.3095±0.0001 | **0.4007±0.0038** | **0.3058±0.0006** | **0.3625±0.0000** |
| Active$^3$ | 0.3109±0.0001 | 0.4464 ±0.0005 | 0.3185±0.0000 | **0.3625 ±0.0000** |
| Passive (ens) | 0.2987 ±0.0001 | 0.2698±0.0000 | 0.2651±0.0001 | **0.3137±0.0002** |
| Active$^1$ (ens) | 0.2911 ±0.0001 | 0.2797±0.0000 | **0.2558±0.0000** | 0.3146±0.0000 |
| Active$^2$ (ens) | 0.2987±0.0001 | 0.2864±0.0003 | 0.2649±0.0001 | 0.3214±0.0005 |
| Active$^3$ (ens) | **0.2935±0.0000** | **0.2665±0.0000** | 0.2615±0.0001 | 0.3221±0.0004 |

**Table 3:** Average test error rates(five trials) on Banknote, MNIST and CIFAR10 for the model selected with 15% unlabelled pool data. Passive/Active stands for the non-ensemble classifiers selected by the $\mathbf{PD}_1$ homological similarities. Passive/Active (ens) stands for the classifiers ensembled from two classifiers: one is selected by the $\mathbf{PD}_1$ homological similarities and the other one is selected by the validation error. The subscript 1, 2 and 3 of the active learning indicates the used 3NN, 5NN and 7NN graphs. Best performance in the non-ensemble and ensemble cases are boldfaced.

## Footnotes

[1]Note that it is conceivable that the decision boundary is not strictly a manifold. While this assumption is critical to the rest of this paper, it is possible to extend the results here by following the theory in [12]. We will leave a thorough exploration of this for future work.

[2] A set $W$ is $\frac{\gamma}{2}$-dense in $\mathcal{M}$ if for every $\mathbf{p} \in \mathcal{M}$ there exists a $\mathbf{x} \in W$ such that $\|\mathbf{p} - \mathbf{x}\|_2 < \frac{\gamma}{2}$. In other words, there exists at least one $\mathbf{x} \in W$ in $B_{\frac{\gamma}{2}}(\mathbf{p})$ for every $\mathbf{p} \in \mathcal{M}$.

[3]The $k$-radius nearest neighbor graph connects all pairs of vertices that are a distance of at most $k$ away, and the $k$-nearest neighbor graph connects a vertex to its $k$ nearest neighbors