[Reviews · NeurIPS 2020]

Review 1

Summary and Contributions: This paper proposes an active learning algorithm to estimate the homology of the decision boundary. The proposed idea is based on the previous work [3] which recovers the homology of the decision boundary. But this paper focuses on the active learning strategy and proves a bound of the necessary samples to fully recover the full LC-complex (which is roughly the Cech complex restricted to the decision boundary, a concept introduced in [3]). An existing active learning method [13] are used to recover the decision boundary. In experiments, the paper shows that model selection can be achieved by comparing the persistence diagrams of the reconstructed LC-complexes reconstructed with the classifier prediction and with the ground truth labels.

Strengths: Checking the homology of the decision boundary is an interesting direction and there can be a lot of potential applications. The theorem seems correct, even though I did not fully read through it. Active learning is certainly the right way to tackle the problem.

Weaknesses: The contribution over [3] is not very surprising, although overall the direction is still quite interesting. I am concerned about how realistic these assumptions are, regarding to the doubling dimension, r-denseness, etc. It will certainly be beneficial to have some empirical estimation of these quantities in the real data. Ideally, the author should consider approximating the bound in theorem 1 in experiments. This way we can know better about how much data is necessary for truthful estimation of the homology. In experiments, instead of the betti number \betta_1, the persistence diagram is used. This leaves some gap that should be carefully addressed. How is the persistence diagram related to the betti number of an (\epsilon, 2r)-LC complex, and how will theorem 1 be extended to the persistence diagrams, which inspects the LC complex with all different radii. In Table 1, validation alone should also be reported to provide a more complete view. -- Post-rebuttal --- I am willing to increase my score to 6. I think the authors addressed my questions well. Overall I appreciate that they are willing to admit the gap in some of the raised issues, although practical performance can be used as indirect evidence. I also read others' reviews and the corresponding rebuttals. I think the issue of the proof raised by R2 can be fixed in the final version without much changing of the proof structure. I think R4 did raised two very valuable questions: density near the decision boundary and whether to actually use topological information for active learning. I think these questions were not addressed well in the rebuttal. This is the reason I am not raising my score to 7.

Correctness: I believe so, although I did not completely check all proofs.

Clarity: yes.

Relation to Prior Work: the difference from [3] is clear.

Reproducibility: Yes

Additional Feedback:


Review 2

Summary and Contributions: This paper proposes an active learning algorithm to recover the homology of decision boundaries, using persistent homology on labeled Cech complexes. The correctness and the effectiveness of the algorithm is theoretically and empirically justified.

Strengths: The performance gain of the method is empirically well verified in the experiments. As far as I know, the idea of considering homology of decision boundaries in active learning setting is new.

Weaknesses: As I mentioned below, the proof of Theorem 1 has some incorrect part and needs to be fixed. As mentioned below, as far as I can think of, this is fixable but requires some work.

Correctness: As far as I checked, the proof of Theorem 1 has some incorrect part and needs to be fixed. In fact, the proof of Theorem 1 (which is Theorem 2 in supplement) is coming from Theorem 3 in [6], and there is a missing condition in Theorem 3 in [6]. When applying Theorem 7.1 in [2], you need to check not only that \bar{x} is r-dense in M, but also that \bar{x} is contained in r-tubular neighborhood of M. Hence in Theorem 3 in [6], we also need to check that \bar{z} is contained in r-tubular neighborhood of M. And this is not implied from the condition of \mathcal{D} being in r-tubular neighborhood of M in page 3 in [6], since the support of the measure mu_z is superset of \mathcal{D} but not necesarily equal. The same problem exists in the paper: Assumption 1 at line 118 assumes that \mathfrak{D} is in r-tubular neighborhood of M, but the observed data \mathcal{D} need not be a subset of \mathfrak{D}, i.e., \partial C \subset Tub_r(M) in line 166. This is because there can be a region where either one of p_{X|Y}(x|1) or p_{X|Y}(x|0) is 1 and the other is 0, i.e., we cannot guarantee that all the samples are coming from \mathfrak{D} even if we restrict to where the edges in E are generated. I discussed how to fix this in additional comments.

Clarity: I think it would be helpful to give the exact definition of the condition number at least in the supplement, for readers who are not familiar with those concepts. In particular, "condition number" has been studied separately in many papers but I think the earliest discovery of this concept was in [Federer, 1959] (it is called reach in this paper). So I think it's worth mentioning Federer's paper. There are two "Assumption 1", one in line 63 and the other in line 118.

Relation to Prior Work: I think the comparison to previous work is well explained in Section 1.

Reproducibility: Yes

Additional Feedback: Fixing the proof of Theorem 1 is not trivial. One can assume that the entire observed data \mathcal{D} is in r-tubular neighborhood of M, but that is out of the point of this paper because we only want the boundary to be in the tubular neighborhood of M. Another possible fix would be as follows: with probability 1, \mathcal{D}^{0} will be a subset of {x:p_{X|Y}(x|0)>0} and \mathcal{D}^{1} will be a subset of {x:p_{X|Y}(x|1)>0}. Hence when we choose an edge {x_i,x_j} in E so that ||x_i - x_j|| < kappa sqrt{rho_i rho_j}, then this edge intersect with both {x:p_{X|Y}(x|0)>0} and {x:p_{X|Y}(x|1)>0}, and hence intersect with the boundary \mathfrak{D} as well. Then the edge {x_i,x_j} is in (kappa sqrt{rho_i rho_j})-tubular neighborhood of \mathfrak{D}, and hence r + (kappa sqrt{rho_i rho_j})-tubular neighborhood of M. And hence the entire edge set E, and eventually the labeled Cech complex, is in r + (kappa sup_i rho_i)-tubular neighborhood of M. The term sup_i rho_i goes to 0 as n goes to infty, so with proper range of kappa so that r + (kappa sup_i rho_i) is bounded by (3-sqrt{8})tau in Assumption 1(b) in line 118 and also that \partial C is r + (kappa sup_i rho_i) - dense in M, then the labeled Cech complex will deformation retracts to M, and hence this will appear in the persistent homology as well. This is very rough sketch and if the authors want to use this proof, they should fill the gap by themselves, although I can help to check more details if the authors want in the authors' response phase. Minor typos: p.2, line 69: "to to" -> "to" p.2, footnote 1: "thge" -> "the" p.2, footnote 1: "work" -> "work." p.3, line 118: "$\mathfrak{D}\in Tub_{r}(\mathcal{M})$" -> "$\mathfrak{D}\subset Tub_{r}(\mathcal{M})$" p.4, line 147: "shortest shortest" -> "shortest" p.7, line 281: "the the" -> "the" Supplement, p.2, line 38: "being draw" -> "being drawn" After reading the authors' response, I agree that will fix the proof of Theorem 1. And hence I increased my score to 6. Although, I would also like to mention that there is a slight mismatch between the framework of the paper and Theorem 1. In line 156, when the graph is constructed, the threshold of the edge is dependent on the k-nn distance at corresponding vertices and uses $\kappa \sqrt{rho_i rho_j}$. However, in Theorem 1, a fixed threshold 2r for the edge is used. Stating and proving the theorem for simpler case is still fine, but I think fixing the theorem so that theorem can match the framework would be better, i.e. providing a version of Theorem 1 for threshold being $\kappa \sqrt{rho_i rho_j}$ would be a good thing to add. And I believe that this kind of extension shouldn't be too difficult using that k-nn distance can be uniformly bounded with high probability.


Review 3

Summary and Contributions: This paper describes a method for analyzing the decision boundaries of data and pre-trained classifiers using tools from topological data analysis. In particular, the method is concerned with picking examples to efficiently learn the homology of data on or near the decision boundary. This is an active learning approach. Not in selecting examples to train the classifier, but in selecting samples to efficiently discover the manifold/decision boundary of the pre-trained classifier. They describe a 2 step process (not iterative, just one method after the other) 1) Use S2 for active learning selection of data points (off the shelf) 2) Construct graph for selected labelled data (bipartite with edges between classes), add more edges (2-hop neighbors) and results in a "simplicial-complex" - another graph. This graph can then be supplied to off-the shelf persistent homology tools to generate "Persistence diagrams" and "Betti numbers" that characterize the topology of the decision boundary (/manifold) They provide theoretical analysis of their method to show that S2 is well suited to pick examples that reveal the topological nature of the manifold efficiently. They then use "homological summaries" to match datasets to models. They claim to be able to select the best model for a given dataset based on comparison between the topological statistics for the dataset and the same statistics for a model (given model outputs)

Strengths: The contribution is straightforward and the results are fair. The stated benefit is that in cases where labels are expensive to obtain, and it is clear that using S2 first would probably help with this situation. It is possible that using topological analysis to give insights into both data samples / manifolds and model decision boundaries might be very helpful. It may be a growth area.

Weaknesses: Method depends on constructing simplexes that are build from nearest neighbor that cross the decision boundary within some radius. Does this mean that the data must be dense at the decision boundary? Would a decision boundary that is furthest from the sample data be more desirable? From the title, I was expecting a method that makes intelligent data point selections by observing the currently discovered topology statistics. Shouldn't the homology results from a few samples guide the active learning from the title? Instead there are 2 distinct phases, one after the other with S2 data point selection having nothing to do with homology analysis (although they do try to show that S2 is suited to helping the other) One stated goal is to enable matching of datasets to pretrained models by comparing topology statistics for both model and dataset in a "model marketplace" where data is matched to pretrained models. see [ref 3]. I am not convinced of the value of this, even without the active learning contribution. Unless the models happen to be trained on the same data I am sure some more fine tuning on the model would be required if so, this should be demonstrated. I also suspect there are far easier statistics to acquire to see if a model "matches" some new data. Eg. what proportion of my data is output positive or negative by my classifier? What is the mean input and output? So I think this method for matching data to models should be compared with other methods that compare other statistics.. With the added constraint that the data should be expensive to require and therefore an active learning approach might be required, I am not sure the approach has many real world applications.

Correctness: These are small datasets, with limited binary classification datasets constructed from MNIST and CIFAR10. Are there problems scaling this up to larger datasets? They do show that using S2 (active) to select samples is better than random (passive) selection when selecting subset of samples to later use in homology analysis in most cases.

Clarity: The setup for the model matching experiments is a little hard to follow, lines 265 to 269 are not clear. What does this sentence mean: "The data pool is prepared for examples query." This is not at all true: Quote From 1.Introduction: "Meta learning refers to a family of algorithms that asses the complexity of the target data to select and appropriate learning model for solving a problem of interest."

Relation to Prior Work: Using topology tools for describing the manifold in terms of PD diagrams and Betti numbers was written about in [ref 3], so the main contribution of this work is the use of S2 to pre-select a subset of data points before applying the methods in ref [3]. This isn't a significant enough breakthrough in itself, so they provide some theoretical analysis. I am not an expert on the theory, they show that "S2 is naturally turned to focusing the labels within Rub_r(M)" which presumably means that S2 picks samples near the manifold, and so the "LC complex" constructed from them can approximate "the homology of the Manifold". This work is said to be the first active learning approach using topological features to learn the homology.

Reproducibility: Yes

Additional Feedback: Minor issues: Quote: "The bottom row of fig 7" : There are no rows in fig 7 Text on axes in figs 6 and 7 is too small to be legible in print out. "shortest shortest path" is somewhere in the text Why is this sentence where it is in fig 6.: "Nonensemble selection by homological similarities" --- post rebuttal -- My overall view of the method as a 2 stage incremental model, and it's utility hasn't changed, but I am unable to follow the proof, whereas other reviews have followed it, and have found it to be a major source of value in the paper. As a result I have kept my overall score but reduced the "confidence score" in my review to the lowest value.

[Author Response · NeurIPS 2020]

We thank the reviewers for their thoughtful feedback and for their appreciation of the novelty of
considering query-efficiency in finding homology of decision boundaries using active learning[1].
**R1: Realism of assumptions and usefulness of upper-bounds.** The assumptions made are known
to give strong indications of the practical applicability of the algorithms and are standard in literature
(see, e.g., the seminal paper [11]). In section 6 of the supplement, we provide numerical complexity
comparison and show that the proposed framework uses *ten times fewer labels* than passive learning
in the example considered. This is also anticipated in the real dataset provided that the conditional
number $1/\tau$ (intrinsic complexity) of the manifold is high (lines 63-68). **R1: Extending Theorem 1**
**to persistence diagrams.** This is an excellent point. Persistence diagrams encode the birth and death
times of topological features as a function of $\epsilon$ of the LC-complex. Our experimental results in Figure
4 and Figure 7(b) of the main paper show that samples found by active learning generate persistence
diagrams closer to ground-truth ones than passive learning. Directly relating our theoretical results to
persistence diagrams is a more fundamental question in manifold learning, and is out of the scope of
the theory of the current manuscript; it is certainly an interesting avenue for future work – we will
remark on this in the final version.
**R2: The containment of $\partial C$ and LČ complex in the tubular neighborhood of $\mathcal{M}$.** We very
much appreciate the R2's detailed review and technical comments. First, we want to clarify a typo
in line 122 where $\mathcal{D}^0$ there actually refers to points of class 0 in the LČ complex but not the entire
dataset. Nevertheless, as R2 points out, assumption 1(a) (line 118) does not guarantee that all samples
of $\partial C$ or the LČ complex falls within the tubular neighborhood of $\mathcal{M}$ (the same issue occurs in [3]).
However, this can be mitigated by requiring a minor extra constraint on the tubular neighborhood of
$\mathfrak{O}$ – under assumption 1(a), we only require that $3r$ is bounded from above by $(\sqrt{9} - \sqrt{8})\tau$. To see
this, let $Tub_{r'}(\mathfrak{O})$ to denote a $r'$-radius tubular neighborhood of $\mathfrak{O}$. For samples in a covering ball
$B_{r/2}(\mathbf{x})$ on manifold $\mathcal{M}$, a k-radius nearest neighbor graph requires $k >= 2r$ to have two furthest
samples of opposite labels connected. That said, after constructing a k-radius nearest neighbor graph
$G$ to satisfy lemma 1 (line 172), the smallest region covered by $\partial C$ of $G$ (line 165) is $\mathfrak{O} + Tub_{2r}(\mathfrak{O})$.
This should guarantee all samples of $\partial C$ come from maximum allowed tubular neighbourhood of $\mathcal{M}$.
We remark here that the same fix applied to the LČ complex will help correct [3]. We will make this
change in final version.
**R3: Importance of density near the decision boundary.** Assumption 1(a)(line 118) indicates
density near decision boundaries is nonzero. As a result, provided sufficient samples from the density,
the proposed framework will succeed; notice that the focus here is on labeling efficiency and not
sample complexity. **R3: Using topology to guide active sample acquisition.** This paper presents
the first analysis of active learning for homology recovery with efficient labeling, and we adopted a
simple but effective two-stage framework. Using the topology statistics to guide active learning is a
very compelling avenue for future work. **R3: On realism of the model marketplace, comparison**
**to other statistics, and other applications.** Our "model marketplace" application is different from
[3]. As R3 suggests, we trained a bank of classifiers with the *same set of training data* (line 290-294)
and verified model selection with the validation data from the same distribution; this is compelling
evidence for the proposed framework, and it could indeed be made stronger by comparison with
other statistics. Currently, our experiments are used to demonstrate that we find the homology of
decision boundaries with fewer labels. Other applications where our work applies can be label
efficient implementation of a topological regularizer [2], complexity measure [6], and finding coresets
that preserve the homology of the decision boundary. In figure 1 below, we show results on the
coreset application in binary classification in MNIST. As observed, predominantly active learning
outperforms passive learning from coresets.

**Figure 1:** Training classifiers with a coreset of 300 data points sampled by active learning/passive learning.
The orange (resp. blue) bars represent the number of MNIST classification cases (out of 45) where active (resp.
passive) learning outperforms the other in a certain range of parameters (line 271-276) of the classifier.

## Footnotes

[1]all references refer to the main manuscript


[Meta-Review · NeurIPS 2020]

The authors presents an application of active learning to the problem of finding the homology of classifier/dataset decision boundaries. The method shows benefits empirically for learning a homology and is also paired with upper bound guarantees on the required number of labels. There was a concern raised around the correctness of the label complexity proof, which the authors have suggested a fix for in the rebuttal. This has satisfied the reviewer, however, there is an additional gap between theory and the algorithm implementation pointed out. This was not deemed a critical flaw, but please do discuss it in the final version. There was also a valid criticism raised regarding the empirical evaluation of using the homology for the purpose of model selection; the investigation would be stronger there were also comparisons to baselines that match models to datasets with simpler heuristics (e.g. matching class distributions). I view the model selection application outside the immediate scope of this paper, nonetheless, a comparison to simpler baselines would strengthen the paper.